# rePIRL: Learn PRM with Inverse RL for LLM Reasoning

Xian Wu [* 1]   Kaijie Zhu [* 2]   Ying Zhang [3]   Lun Wang [4]   Wenbo Guo [2]

## Abstract

Process rewards have been widely used in deep reinforcement learning to improve training efficiency, reduce variance, and prevent reward hacking. In LLM reasoning, existing works also explore various solutions for learning effective process reward models (PRM) with or without the help of an expert policy. However, existing methods either rely on strong assumptions about the expert policies (*e.g.,* requiring their reward functions) or suffer intrinsic limitations (*e.g.,* entropy collapse), resulting in weak PRMs or limited generalizability. In this paper, we introduce rePIRL, an inverse RL-inspired framework that learns effective PRMs with minimal assumptions about expert policies. Specifically, we design a dual learning process that updates the policy and the PRM interchangeably. Our learning algorithm has customized techniques to address the challenges of scaling traditional inverse RL to LLMs. We theoretically show that our proposed learning framework can unify both online and offline PRM learning methods, justifying that rePIRL can learn PRMs with minimal assumptions. Empirical evaluations on standardized math and coding reasoning datasets demonstrate the effectiveness of rePIRL over existing methods. We further show the application of our trained PRM in test-time training, test-time scaling, and providing an early signal for training hard problems. Finally, we validate our training recipe and key design choices via a detailed ablation study. Code is available at https://github.com/ucsb-mlsec/irl.

## 1. Introduction

Intermediate rewards have demonstrated their effectiveness in training deep reinforcement learning agents for various applications, *e.g.,* simulated games (Ng et al., 1999), Go (Silver et al., 2016), and Poker games (Brown & Sandholm, 2019). For example, in MuJoCo environments, intermediate rewards are introduced to guide the robot to progressively learn to stand, walk, and finish its tasks (Heess et al., 2017). Intermediate rewards are also helpful in accelerating learning convergences (Wu et al., 2023), reducing variances (Laud, 2004), and preventing reward hacking (Glaese et al., 2022). Inspired by these successes, recent works also explore training process reward models (PRMs) to provide token-level intermediate reward during LLM reasoning. At a high level, these works can be categorized as offline methods (Uesato et al., 2022; Lightman et al., 2023; Wu et al., 2023; Xu et al., 2024; Yoon et al., 2024), which learn a PRM based on a given expert policy, or online methods (Wang et al., 2024; Zhao et al., 2025b; Cui et al., 2025a), which learn a PRM based on internal signals of the reasoning model or the final outcome reward. However, existing PRM methods either add strong assumptions about the expert policies and reward modeling or suffer intrinsic limitations.

Specifically, within offline methods, supervised PRM methods (Uesato et al., 2022; Lightman et al., 2023) or DPO-type of works (Yuan et al., 2024; Cui et al., 2025a) (although do not explicitly learn a PRM) require access to token-level annotations or preference labels of expert trajectories. Monte Carlo tree search (MCTS)-based methods require running the expert policy and recording the final outcome reward, which is time-consuming and no longer works if the expert policy is not available. For online methods, PRIME (Cui et al., 2025a) learns a PRM based on the outcome reward of the current policy, which only works under strict assumptions about the reward functions and the policy model under training (see Section 3.3). Intrinsic reward methods (Zhao et al., 2025b) explore using the reasoning model's response entropy or confidence as the process reward, but suffer from the entropy collapse issue, degrading performance in later training stages (Cui et al., 2025b).

In this paper, we propose rePIRL, a novel learning framework that learns effective PRMs for LLM reasoning from only the expert trajectories without access to process reward

---

[*]Equal contribution [1]Meta AI [2]Department of Computer Science, University of California, Santa Barbara [3]Independent Researcher [4]Google DeepMind. Correspondence to: Xian Wu <xianwu123@meta.com>, Wenbo Guo <henrygwb@ucsb.edu>.

*Proceedings of the 43rd International Conference on Machine Learning*, Seoul, South Korea. PMLR 306, 2026. Copyright 2026 by the author(s).

annotations, preference labels, and the expert policy. rePIRL also does not require the strong assumptions of the online method, PRIME, and does not suffer the entropy collapse issue of the intrinsic reward methods. At a high level, we borrow the idea from inverse reinforcement learning (IRL), which is proposed to learn a reward function from given expert trajectories, using which one can learn a policy as optimal as the expert policy. We start with modeling multi-step LLM reasoning as a token-level MDP and design the learning objective function for our PRM following the classical IRL framework, *i.e.,* defining a hidden variable followed by energy-based modeling. To make sure the objective function is tractable for LLMs, we then apply the gradient trick to get rid of the partition function and apply importance sampling to avoid sampling from the expert policy (which is not available). Following this process, we design our learning algorithm to learn PRM and policy interchangeably, where the policy loss is the maximum entropy RL loss, which can be solved by PPO (Schulman et al., 2017) or its variants (RLOO (Ahmadian et al., 2024), GRPO (Shao et al., 2024)). After deriving our learning algorithm, we further conduct rigorous theoretical analysis to integrate representative online and offline PRM learning methods into our framework: PRIME (Cui et al., 2025a), Math-Shepherd (MCTS) (Wang et al., 2024), DPO (Rafailov et al., 2023), and DQO (Ji et al., 2024). We show that these methods can be integrated into our learning framework with additional assumptions on top of rePIRL, justifying that *rePIRL is the PRM learning method with the minimal required assumptions*.

We compare rePIRL with a set of baselines using Qwen2.5-3B and Qwen3-4B as base models on seven standard math and coding reasoning benchmarks. Our results show the superiority of our method compared to offline and online PRM baselines. We then demonstrate the utility of our trained PRM in three scenarios: testing-time training, where rePIRL matches verifiable outcome rewards methods, testing-time scaling, where rePIRL can be used to select better rollouts, and hard-problem only training, where rePIRL can provide useful early signals for training difficult problems. Finally, we conduct a detailed ablation study to validate our training recipe, including base model selection, data selection, and RL algorithms. To the best of our knowledge, rePIRL is *the first work that generalizes IRL for LLM reasoning*. It is also *the first work that unified the existing PRM learning methods into the same learning framework*.

Our key contributions are: 1) We propose rePIRL, a novel PRM learning framework, inspired by classical IRL, together with a scalable learning algorithm for both PRM and policy training. 2) We proposed a unified post-training framework that includes SOTA reasoning methods, especially DPO, DQO, PRIME, and MCTS. For each method, we analyzed its underlying assumptions and connections to our framework. Through this analysis, we establish a unified

framework, a better understanding of different reasoning methods, and a conclusion that rePIRL is a generalized PRM learning method that operates under the minimal necessary assumptions. 3) We empirically demonstrate the clear advantage of rePIRL over baseline methods and the utility of the trained PRM in various scenarios.

## 2. Related Work

There have been a number of works on LLM reasoning that learn process reward models with or without expert knowledge. As shown in Table 1, these methods generally adopt stronger assumptions than ours, and can be integrated into our unified framework outlined in Section 3. We group the prior research by whether it requires expert policies.

**Methods require expert policies.** Early works in this category train a process reward model from data with annotated token-level or step-level rewards using supervised learning losses (Uesato et al., 2022; Wu et al., 2023; Xu et al., 2024; Yoon et al., 2024; Ma et al., 2023; Lightman et al., 2023). The annotations are given by human experts or other well-trained LLMs. Without assessing the token-level rewards, another line of work proposes to run the expert policy-based Markov chain tree search and generate trajectories with process reward labels (Zhao et al., 2025a; Lee et al., 2024; Zhang et al., 2024b; Wang et al., 2024; Zhang et al., 2024a; Xiong et al., 2025b). Here, the label for each intermediate reasoning step is assigned based on the final outcomes of the Markov samples drawn from this step. These methods require access to the expert policy. More importantly, the MCTS process is time-consuming and computationally expensive, especially when calculating token-level rewards.

Note that directed policy optimization (DPO) and its followups (Rafailov et al., 2023; Richemond et al., 2024; Ethayarajh et al., 2024a) directly learn new policies from requiring expert trajectories without explicitly modeling the reward function. DPO has the bandit assumption, *i.e.,* the model cannot consider multi-step reasoning. Recent offline RL works extend DPO to multi-step reasoning by leveraging a trajectory-level Bradley-Terry preference model (Rafailov et al., 2024; Zeng et al., 2024; Meng et al., 2024) or leveraging the relations between the optimal policy model and value functions in maximum entropy RL (Ji et al., 2024; Wang et al., 2025a; Zhong et al., 2024). As these works do not explicitly learn a reward function, we do not consider them as baselines in our work.

**Methods do not require external signals.** Yuan et al. (2024) proposes a specific format for the reward function, which can be used as both process and outcome rewards. Cui et al. (2025a) (PRIME) leverages this reward format and proposes to learn a PRM based on outcome reward and policy interchangeability. As we will show in Section 3,

*Table 1.* Comparison of rePIRL and SOTA PRM methods. ✓ means the corresponding method does not suffer this issue, ✗ otherwise. "Supervised" refers to the methods that learn PRMs from supervised labels and "Intrinsic" refers to the methods that learn PRMs from model confidence or entropy. Assumptions refer to the method-specific assumptions discussed in Section 3.3.

|  | Supervised | MCTS | DPO | PRIME | Intrinsic | rePIRL |
|---|---|---|---|---|---|---|
| Token annotation/label | ✗ | ✓ | ✗ | ✓ | ✓ | ✓ |
| Expert reward | ✗ | ✗ | ✗ | ✓ | ✓ | ✓ |
| Expert policy | ✗ | ✗ | ✗ | ✓ | ✓ | ✓ |
| Assumptions | ✓ | ✗ | ✗ | ✗ | ✓ | ✓ |
| Entropy collapse | ✓ | ✓ | ✓ | ✓ | ✗ | ✓ |

this specific format only holds under certain assumptions (Section 3.3). Other works explore using model internal confidence or entropy as process reward (Zhao et al., 2025b; Li et al., 2025a). While such approaches can be beneficial during the early stages of fine-tuning, they often suffer from entropy collapse, which can degrade performance in later training stages (Cui et al., 2025b).

Note that we do not consider the works that learn reasoning models through supervised imitation learning (distillation) (Singh et al., 2023; Dong et al., 2023; Hao et al., 2024; Muennighoff et al., 2025; Li et al., 2025b; Ye et al., 2025; Xia et al., 2025; Kang et al., 2025; Wang et al., 2025b; Cui et al., 2025c; Ma et al., 2025). Some recent methods improve the online RL algorithms (*e.g.,* PPO) for LLM reasoning (Xiong et al., 2023; 2025a; Arora & Zanette, 2025; Guo et al., 2025; Luong et al., 2024; Yeo et al., 2025; Yu et al., 2025; Yuan et al., 2025; Hu, 2025; Li et al., 2023; Williams, 1992; Ahmadian et al., 2024; Schulman et al., 2017), which can be used in our methods to update the policy model. Finally, there are some studies of inverse RL before the emergence of LLMs (Ziebart et al., 2008; Finn et al., 2016b). There are also some works about inverse RL for alignment (Sun & van der Schaar, 2024; Zeng et al., 2025; Li et al., 2024b). These works are designed for alignment rather than reasoning. As such, the fundamental model is different. As stated in their papers, alignment mainly optimizes over the entire trajectories, which fundamentally follows a bandit assumption. However, reasoning requires modeling the problem as a real MDP, making the problem much more challenging.

## 3. Key Technique

### 3.1. Problem Setup

Given a reasoning LLM denoted by $f$, which takes as input a prompt $\mathbf{x}$ and generates the output tokens autoregressively, where each token is denoted as $y_i$. We define the generation process as a token-level Markov decision process (MDP): $\mathcal{M} = \{\mathcal{S}, \mathcal{A}, T, R, \gamma\}$, where state $s_t \in \mathcal{S}$ is the current output token $s_t = [\mathbf{x}, y_0, ..., y_{t-1}]$. $\mathcal{A}$ is the vocabulary size, and $a_t = y_t$ is the LLM's current output token at time/position $t$. As the next state is obtained by appending

the newly generated token to the current output, we have a deterministic state transition function (T). $R(s_t, a_t)$ is the reward for each generated token. For simplicity, we omit the discount factor $\gamma$. Given a set of trajectories $\mathcal{D}$ sampled from an expert policy $\pi_{\text{ref}}$, where each trajectory $\tau = \{(s_t, a_t)\}_{t=1:T}$. The expert policy is trained from a certain reward function $r_{\text{ref}}(s, a)$. As shown in Table 1, we consider the setup with the fewest assumptions, where we do not assume accessing the reference policy $\pi_{\text{ref}}$ and its reward function $r_{\text{ref}}(s, a)$. For the expert trajectories, we do not require the token-level reward or preference labels. Our *goal* is to recover a reward function from the same family as the reference reward function $r_{\text{ref}}(s, a)$ and recover a policy that is as optimal as possible under the recovered reward model.

First, having the fewest assumptions reduces the thresholds of applying our proposed method, making it much more generalizable. For example, obtaining token-level reward labels requires either substantial human effort or intensive computational cost to conduct MCTS. Our method can operate without such additional efforts and other assumptions. Besides, because our method does not assume access to an expert policy, it can learn from trajectories sampled from experts that do not have an analytic policy function (*e.g.,* humans). Although it is possible to directly learn a nearly optimal policy from the expert trajectories without explicitly learning the reward function (Wang et al., 2025a), we believe it is still necessary to recover the reward function in many scenarios. This is because the learned reward function can be used together with other reward functions for further training the policy to learn new capabilities and improve its end-to-end performance. For example, if the reference policy often produces long reasoning processes with redundant steps (*e.g.,* DeepSeek-r1), we can add a reward that penalizes the long reasoning chain. By using it together with the recovered reward, we can potentially train policies that can produce concise reasoning chains without harming their final performance. Another use case is when the expert and the learned policy cannot handle a specific type of query well (could be unseen questions), we can use the recovered reward and a reward function designed specifically for those queries to train the policy such that it can better handle these queries while preserving its general performance.

## 3.2. rePIRL Learning Framework

**Technical rationale.** We borrow the idea from classical inverse reinforcement learning, as these techniques are designed for learning a reward function from given expert trajectories without the reward annotations. Following this idea, a straightforward solution would be using an inverse RL method to learn a reward function from the expert trajectories and then train a policy based on the learned reward function. However, the traditional inverse RL problem requires estimating a partition function $z$, which involves traversing all the possible states and actions (Ziebart et al., 2008). This is intractable for LLM models, which have a large action and almost infinite state space.

An approximate solution is to train a policy model together with the reward function and approximate the partition function using the trajectories sampled from the trained policy model. This method avoids the intractable operations in the traditional inverse RL models and has been demonstrated to be effective for shallow neural network policies in robotics (Finn et al., 2016b). In this work, we generalize this solution to LLM multi-step reasoning and propose a unified reward and policy learning framework.

**Objective function construction.** We define the reward function as $r_\phi(s_t, a_t)$, parametrized by $\phi$ and the policy network as $\pi_\theta(a_t|s_t)$. We then define a hidden variable $o_t = 0/1$. $o_t = 1$ means the action $a_t$ in an expert trajectory is sampled from the expert policy rather than a non-optimal one, and $o_t = 0$ otherwise. We define $p(o_t|s_t, a_t) \propto \exp(r_\phi(s_t, a_t))$, an energy-based function (Finn et al., 2016a). Here, $p(\tau|o_{1:T})$ is the probability of $\tau$ being sampled from the expert policy.

Reward function. We can learn the reward function from the following MLE loss: $\mathcal{J}(\phi) = \mathbb{E}_{\tau \sim \mathcal{D}}[\log p(\tau|o_{1:T})]$. That is, given a set of trajectories sampled from the expert policy, we want to maximize their likelihood by learning an optimal reward function.

$$\max_\phi \mathcal{J}(\phi) = \max_\phi \frac{1}{|\mathcal{D}|} \sum_i \sum_t r_\phi(s_{i,t}, a_{i,t}) - \log z(\phi),$$
(1)

where $s_{i,t}$ is the $t$-th state in the $i$-th trajectory ($\tau_i$). $z(\phi) = \int p(\tau) \exp(r_\phi(\tau)) d\tau$ is the partition function. The gradient of the loss function w.r.t. the reward function parameter is

$$\nabla_\phi \mathcal{J} = \mathbb{E}_{\tau \sim \mathcal{D}}[\nabla_\phi r_\phi(\tau)] - \mathbb{E}_{\tau \sim p(\tau|o_{1:T})}[\nabla_\phi r_\phi(\tau)],$$
(2)

where $p(\tau|o_{1:T}) = \frac{p(\tau) \exp(r_\phi)}{z(\phi)}$, which is the soft optimal policy under the current reward. Estimating $p(\tau|o_{1:T})$ requires computing the forward and backward pass $\mu_t(s_t, a_t) \approx \beta(s_t, a_t)\alpha(s_t)$ for all time steps (Ziebart et al., 2008). For the token-level MDP, $\beta(s_t) = \mathbb{E}_{a_t \sim p(a_t|s_t)}[\exp(r_\phi(s_t, a_t))\beta(s_{t+1})]$ and $\alpha(s_t) = \sum_{a_{t-1}} \exp(r_\phi(s_{t-1}, a_{t-1}))\alpha(s_{t-1})$. Because the

state and action spaces are enormous and the reasoning models can span many time steps, computing $\alpha$ and $\beta$ becomes intractable.

To resolve this challenge, we propose to sample trajectories $\bar{\tau}$ from our defined policy $\pi_\theta$ rather than the $p(\tau|o_{1:T})$. The gradient in Eqn. (2) is then turned into

$$\nabla_\phi \mathcal{J} = \mathbb{E}_{\tau \sim \mathcal{D}}[\nabla_\phi r_\phi(\tau)] - \mathbb{E}_{\bar{\tau} \sim \pi_\theta(a_t|s_t)}\left[\frac{p(\bar{\tau}|o_{1:T})}{\pi_\theta(\bar{\tau})} \nabla_\phi r_\phi(\bar{\tau})\right]$$
$$= \frac{1}{N} \sum_i \nabla_\phi r_\phi(\tau_i) - \frac{1}{\sum_j w_j} \sum_j w_j \nabla_\phi r_\phi(\bar{\tau}_j),$$
(3)

where $\nabla_\phi r_\phi(\tau_i) = \sum_t \nabla_\phi r_\phi(s_{i,t}, a_{i,t})$.

$$w_j = \frac{p(s_1) \prod_t p(s_{t+1}|s_t, a_t) \exp(r_\phi(s_t, a_t))}{p(s_1) \prod_t p(s_{t+1}|s_t, a_t)\pi_\theta(a_t|s_t)}$$
$$= \frac{\exp(\sum_t r_\phi(s_t, a_t))}{\prod_t \pi_\theta(a_t|s_t)} = \frac{\exp(r_\phi(\bar{\tau}_j))}{\pi_\theta(\bar{\tau}_j)}$$
(4)

is the importance weight. Instead of solving the optimal policy under the current reward function, we sample from a sub-optimal policy $\pi_\theta$ and mitigate the sample bias with importance sampling. The more we optimize the policy $\pi_\theta$, the closer the importance weight will go to 1. The final objective function for $r_\phi$ can be written as

$$\max E_{\tau \sim \mathcal{D}}[r_\phi(\tau))] - \log(E_{\bar{\tau} \sim \pi_\theta}\left[\frac{\exp(r_\phi(\bar{\tau}))}{\pi_\theta(\bar{\tau})}\right]). \quad (5)$$

Policy model. At any time during the training, we can update the policy network by the common online RL objective, *i.e.,* maximize the expected total reward $\mathbb{E}_{\bar{\tau} \sim \pi_\theta}[r_\phi(\bar{\tau})]$. However, the optimal policy under this objective function will be a deterministic policy that only takes the action maximizing the value function at each state $a_t = \arg\max Q(s_t, a_t)$. For LLMs, this may cause the degeneration or model collapse issue. To resolve this issue, we propose to learn the policy model via the maximum entropy RL objective function (Haarnoja et al., 2018).

$$\max E_{\bar{\tau} \sim \pi_\theta}[r_\phi(\bar{\tau})] + \beta \mathcal{H}(\pi_\theta(\tau))$$
$$= E_{\bar{\tau} \sim \pi_\theta}[r_\phi(\bar{\tau})] - \beta E_{\bar{\tau} \sim \pi_\theta}[\log \pi_\theta(\bar{\tau})].$$
(6)

**Learning algorithm.** Algorithm 1 shows our proposed learning algorithm for the reward function and the policy model. Specifically, to mitigate numerical instability, we do not compute $w$ directly, Instead, we perform all calculations in log-space: $\log w = r_\phi(\tau) - \sum_{t=1}^{|\tau|} \log \pi_\theta(a_t|s_t)$. The final normalized weights are then derived using a Softmax transformation across the mini-batch. In addition, to mitigate reward hacking, we use the average process reward, rather than the raw cumulative reward, when computing the loss for the process reward model.

---

**Algorithm 1** rePIRL Algorithm for one iteration

---

**Require:** Policy model $\pi_\theta$, token-level process reward model $\pi_\phi$, outcome reward model $r_o$, training set $\mathcal{D}$, rollout number $n$, expert rollouts $\tau_\mathcal{D}$ for training set $\mathcal{D}$.

1: **for** Batch $\mathcal{B}$ in $\mathcal{D}$ **do**
2:      $\hat{\tau}_\mathcal{B} = \pi_\theta(\mathcal{B}, n)$ {Generating $n$ policy rollouts for each question in the batch: $\hat{\tau}_\mathcal{B}$}
3:      Fetch expert demonstration $\tau_\mathcal{B}$ w.r.t batch $\mathcal{B}$
4:      Combine policy rollouts and expert rollout
5:      Calculating the outcome reward $r_o(\tau)$ for each rollout $\tau$
6:      Calculating the average process reward for each rollout: $\bar{r}_\phi(\tau) = \frac{1}{|\tau|} \sum r_\phi(\tau)$
7:      For each rollout, if the outcome reward $r_o(\tau)$ is 0, calculate the rollout importance score $w$:

$$w = \frac{\exp(r_\phi(\tau))}{\pi_\theta(\tau)}$$

8:      Calculate the loss for the process reward model:

$$\mathcal{L} = \frac{1}{\sum w_i} \sum_{\hat{\tau} \in \hat{\tau}_\mathcal{B}} w_i \bar{r}_\phi(\hat{\tau}) - \frac{1}{|\mathcal{B}| * n} \sum_{\tau \in \tau_\mathcal{B}} \bar{r}_\phi(\tau)$$

9:      Update the reward model according to $\mathcal{L}$
10:     Calculate the process reward score for every policy rollout $\hat{\tau}$ in $\hat{\tau}_\mathcal{B}$
11:     Update the policy model $\pi_\theta$ using RLOO algorithm
12: **end for**

---

### 3.3. Integrate SOTA Methods into our Framework

Here, we discuss how existing offline RL and PRM learning methods can be integrated into our proposed framework by adding assumptions with different strengths. We mainly discuss two representative methods listed in Table 1: Math-Shepherd (Wang et al., 2024), which learns a reward function solely, and PRIME (Cui et al., 2025a), which learns a reward function together with the policy model, as well as two methods that learn a policy from the expert trajectories without explicitly learning a reward function: DPO (Rafailov et al., 2024) and DQO (Ji et al., 2024).

First, we present the following theorem.

**Theorem 1** *The optimal solution for the maximize entropy RL objective in Eqn. (6) is $\pi_*(a|s) = exp(\frac{1}{\beta}[Q^*(s, a) - V^*(s)])$, where $V^\pi(s_t) = \mathbb{E}_{a_t \sim \pi(\cdot|s_t)}[r(s_t, a_t) + \gamma V^\pi(s_{t+1})] + \beta \mathcal{H}(\pi(\cdot|s_t))$ is the soft state-value function and $Q^\pi(s_t, a_t) = r(s_t, a_t) + \gamma V^\pi(s_{t+1})$ is the soft action-value function.*

Theorem 1 serves as the foundation for our theoretical analysis. We note that this theorem is an established result and

not a novel contribution of this work. Our novelty is the analysis of different methods' connection to our framework and their assumptions, as follows.

**Connection to DPO.** As shown in Section 2, when being used to multi-step LLM reasoning, DPO assumes the availability of trajectory preference labels and a reference policy where the offline trajectories are sampled from. Given that DPO does not explicitly learn the reward function, it only has one learning objective function for the policy model. The DPO objective is constructed based on the following relations $\pi_*(a|s) = \frac{1}{Z(s)} \pi_{\text{ref}}(a|s) \exp(\frac{1}{\beta} r(a, s))$. The following proposition states this key relations and our policy model objective in Eqn. 6.

**Proposition 1** *When define the reward function as $r'(a, s) = r(a, s) + \beta \log \pi_{ref}$ and the bandit assumption. The optimal solution of the maximum entropy RL is $\pi_*(a|s) = \frac{1}{Z(s)} \pi_{ref}(a|s) exp(\frac{1}{\beta} r(a, s))$.*

This proposition states that the policy learning objective of DPO is a special case of our policy learning objective. The availability of preference label enables DPO to directly learn the policy model using an MLE loss based on the Bradley-Terry model (Bradley & Terry, 1952), eliminating the need to estimate the partition function. However, due to the lack of the partition function, we cannot directly obtain the reward function from a learned policy as $r(s, a) = \beta \log \frac{\pi}{\pi_{\text{ref}}} + \beta logZ$. Note that $\log \frac{\pi}{\pi_{\text{ref}}}$ is consistent with $r(s, a)$ only under the Bradley-Terry model, however, when being used for policy training, it is biased without the partition function. In summary, *under a specific reward function, DPO optimizes the same policy objective as us. It does not need to estimate the reward function and the partition function thanks to the preference label.*

**Connection to DQO.** DQO assumes the access to token-level reward annotation and design their policy objective functions accordingly.

**Proposition 2** *The objective function of DQO is derived from our policy model objective based on the relation in Theorem 1.*

With proposition 2, we can state that DQO *also optimizes the same maximum entropy RL for the policy model*. With the token-level reward annotations, DQO can approximate the value function via TD learning without learning the reward function.

**Connection to Math-shepherd.** Math-shepherd and follow-up works apply MCTS to obtain the process reward annotation and then leverage an MLE (cross-entropy) loss to learn a reward function. The MCTS process is computationally cost and the resulted process reward is a distribution of the final reward and cannot capture the intermediate rewards. In LLM reasoning, the intermediate reward could be the

quality and conciseness of intermediate reasoning steps. Besides, it requires accessing and executing the expert policy. Here, we show that our reward objective in Eqn. (5) can also be modeled as a cross-entropy loss.

**Proposition 3** *The objective function in Eqn. (5) is equivalent to the following cross-entropy loss.* $\mathbb{E}_{\tau \sim \mathcal{D}}[\log D(\tau)] + \mathbb{E}_{\tau \sim \pi_\theta}[\log(1 - D(\tau))]$ *where* $D = \frac{\tilde{p}}{\tilde{p}+\pi_\theta}$ *and* $\tilde{p} = \frac{1}{z}\exp(r_\phi(\cdot))$.

**Connection to PRIME.** Although PRIME is an online method (*i.e.,* does not require expert trajectories), it also learns a reward function together with the policy network. Here we show that PRIME can also be integrated into our framework under certain assumptions. First, PRIME leverages the implicit reward, *i.e.,* model the reward as $r = \log \frac{\pi_\theta}{\pi_{\text{ref}}}$ and use it as the PRM and ORM at the same time. The following proposition states the underline assumptions of having this relationship.

**Proposition 4** $r = \log \frac{\pi_\theta}{\pi_{\text{ref}}}$ *can be used as PRM and ORM at the same time only if the following assumptions are satisfied: 1) The outcome reward is the total reward of the entire episode; 2)* $V(s_t) - V(s_{t-1}) = r(s_t, a_t)$.

These are strong assumptions especially the first one. For example, if the given outcome reward annotation is just whether the final output is correct or not, this method actually just distributes this outcome reward to each step without considering any other intermediate reward.

Furthermore, the reward function objective of PRIME is a cross-entropy loss, which is equivalent to a simplified version of our method without the importance sampling weight, $\max E_{\tau \sim \pi_{\text{ref}}}[\exp(r_\phi(\tau))] - E_{\tau \sim \pi_\theta}[\exp(r_\phi(\tau))]$. For the policy training, PRIME uses a SOTA policy gradient method, which is our policy objective without the entropy constraint. As such, PRIME is a simplified version of our method with extra assumptions.

In summary, all the discussed techniques can be integrated into our proposed framework by introducing additional assumptions. In contrast, our framework makes the fewest assumptions and is therefore the most general, but also the most complicated learning process. The proof of all theorem and propositions can be found in the Appendix A.

## 4. Evaluation

### 4.1. Experiment Setup

**Tasks and datasets.** We assess the reasoning capabilities of our proposed method on seven standard benchmarks spanning competition-level math and coding problems. For mathematics, we use AIME-2024 (Li et al., 2024a), AMC (Li et al., 2024a), Math-500 (Hendrycks et al., 2021), Minerva-

Math (Lewkowycz et al., 2022), and Olympiad-Bench (He et al., 2024); for coding, we use Leetcode (Guo et al., 2024) and LiveCodeBench (Jain et al., 2024). We evaluate all models, including baselines, using greedy decoding and report the zero-shot pass@1 accuracy—the percentage of problems solved correctly on the first attempt. Our training data is constructed by sampling 7,000 instances per domain (*i.e.,* math and coding) from the PRIME Eurus-2-RL-Data (Cui et al., 2025a) dataset. For each sampled problem, we generate four expert demonstration trajectories using Claude-3.7-sonnet (Anthropic, 2025), which serve as the expert data for inverse RL. The validation set uses all 2,050 available samples from the PRIME Eurus-2-RL-Data validation set for model selection.

**Baseline.** We select seven baselines. The first is behavioral cloning (BC) on the expert dataset, a standard benchmark for inverse RL. The second is KTO (Ethayarajh et al., 2024b), a SOTA offline RL method for multi-step reasoning, which learns PRM implicitly. The other three are representative online RL-based PRM training methods: PRIME (Cui et al., 2025a), which derives implicit rewards via a DPO-like loss; MCTS (Wang et al., 2024), which estimates intermediate rewards at each step by rolling out the expert policy multiple times; and RL Tango (Zha et al., 2025), which trains an auxiliary LLM to predict step-wise scores. Finally, we also compare rePIRL with the vanilla RLOO algorithm with outcome rewards, as well as a method that conducts SFT warm-up with expert trajectories followed by RLOO with outcome reward (denoted by SFT+RLOO).

**Model and training setup.** We select *Qwen2.5-3B-Instruct* (Yang et al., 2024) and *Qwen3-4B-Base* (Yang et al., 2025) as our base models, chosen for their strong instruction-following and reasoning abilities, as well as their training efficiency. Our training process consists of two sequential stages. *First*, we fine-tune the base model on mathematical datasets and evaluate its performance on math benchmarks. *Second*, this math-specialized model is used to initialize training on coding datasets, followed by a final evaluation on coding benchmarks. A separate PRM is also initialized from the same base model with an additional head. All experiments were conducted using VeRL (Sheng et al., 2025) on eight RTX PRO 6000 GPUs.

We train both the policy and reward models using AdamW (Loshchilov & Hutter, 2017) with a microbatch size of 1 and minibatch size of 32. The policy model uses a constant learning rate of $5 \times 10^{-7}$ with batch size 128, while the PRM uses $3 \times 10^{-8}$, also with a batch size 128. Policy training follows RLOO with four rollouts per prompt, and the final reward is a weighted combination of outcome and PRM rewards, using a ratio of $1 : 0.05$ for *Qwen2.5-3B-Instruct* and $1 : 0.1$ for *Qwen3-4B-Base*. We also include a policy entropy loss with a coefficient $0.001$. All models

*Table 2.* **Performance comparison with prior methods on mathematical and coding benchmarks.** Our method outperforms baselines on Qwen2.5-3B and Qwen3-4B across both domains. RL models are trained for two epochs, with the best selected via validation. Deterministic evaluation follows vLLM Contributors (2025).

| Model | Math | | | | | | Coding | | |
|---|---|---|---|---|---|---|---|---|---|
| | MATH500 | AIME2024 | MinervaMath | AMC | OlympiadBench | Avg. | Leetcode | LiveCodeBench | Avg. |
| Qwen2.5-3B-Instruct | 46.0 | 10.0 | 22.4 | 34.9 | 28.9 | 28.4 | 26.7 | 18.3 | 22.5 |
| BC | 52.0 | 3.3 | 8.8 | 22.8 | 15.9 | 20.6 | 28.3 | 14.5 | 21.4 |
| KTO | 54.5 | 4.8 | 18.5 | 29.5 | 24.2 | 26.3 | 28.9 | 16.8 | 22.9 |
| RL-Tango | 57.1 | 6.6 | 25.2 | 32.5 | 29.5 | 30.2 | 30.8 | 20.3 | 25.6 |
| MCTS | 58.0 | 6.6 | 24.3 | 33.7 | 30.7 | 30.7 | 31.1 | 20.2 | 25.7 |
| PRIME | 56.2 | 6.6 | 26.5 | 31.3 | 29.0 | 29.9 | 30.0 | 20.4 | 25.2 |
| RLOO | 63.6 | 3.3 | 26.1 | 36.1 | 29.6 | 31.7 | 28.9 | 19.4 | 24.1 |
| SFT+RLOO | 54.0 | 7.0 | 21.3 | 22.9 | 23.3 | 25.7 | 25.0 | 18.1 | 21.6 |
| **rePIRL** | 62.4 | 10.0 | 27.2 | 38.5 | 29.3 | **33.5** | 35.0 | 20.3 | **27.7** |
| Qwen3-4B-Base | 58.6 | 6.0 | 19.9 | 36.1 | 32.0 | 30.5 | 41.1 | 27.0 | 34.0 |
| BC | 60.0 | 4.0 | 14.5 | 28.0 | 22.0 | 25.7 | 38.5 | 24.5 | 31.5 |
| KTO | 66.0 | 5.5 | 24.0 | 36.5 | 34.5 | 33.3 | 42.5 | 26.5 | 34.5 |
| RL-Tango | 71.5 | 8.0 | 30.5 | 40.5 | 38.5 | 37.8 | 45.0 | 23.5 | 34.3 |
| MCTS | 70.6 | 6.0 | 29.0 | 37.3 | 37.6 | 36.1 | 45.5 | 27.4 | 36.4 |
| PRIME | 72.4 | 10.0 | 32.4 | 43.3 | 38.4 | 39.3 | 43.3 | 19.0 | 31.1 |
| RLOO | 73.0 | 10.0 | 29.8 | 39.7 | 39.7 | 38.4 | 46.1 | 24.9 | 35.5 |
| SFT+RLOO | 69.8 | 4.0 | 22.8 | 39.8 | 36.9 | 34.7 | 41.7 | 28.8 | 35.3 |
| **rePIRL** | 72.8 | 16.0 | 33.5 | 43.3 | 42.5 | **41.6** | 52.2 | 27.5 | **39.8** |

are trained for 2 epochs, and the best checkpoint is selected based on validation performance. For baseline models, we adopt default hyperparameters whenever available; otherwise, we match those of our method to ensure fairness. Specifically, for the MCTS baseline, we use the PRM from Math-Shepherd (Wang et al., 2024). Additional training details are provided in the Appendix B.

### 4.2. Main Experiments

Table 2 shows the comparisons between rePIRL and seven baselines on selected benchmarks under greedy decoding. First, rePIRL achieves state-of-the-art performance among 3B- and 4B-scale reasoning LLMs, reaching 33.5% (math) and 27.7% (coding) with *Qwen2.5-3B-Instruct*, and 41.6% (math) and 39.8% (coding) with *Qwen3-4B-Base*. In addition, we can further observe that offline methods (*i.e.,* BC or KTO) underperform other approaches, highlighting the importance of online policy updates if the environment is available. Second, while the online baselines improve over the base Qwen models, they still perform worse than our approach. This result validates the effectiveness of our generated process reward in helping with training better policies.

Finally, compared to the vanilla RLOO algorithm, our method achieves absolute improvements of 2.8% in math and 3.6% in coding with *Qwen2.5-3B-Instruct*, and 3.2% in math and 4.3% in coding with *Qwen3-4B-Base*, demonstrating that our process rewards provide informative guidance.

The superior performance of rePIRL against SFT+RLOO further shows the advantage of IRL-based PRM over SFT warm-up in leveraging expert information. Moreover, We evaluate rePIRL against strong baselines (*i.e.,* RLOO and PRIME) on mathematical benchmarks using pass@1 under non-greedy decoding (Chen et al., 2021) in Appendix C. The results further confirm the superiority of rePIRL over baseline methods.

### 4.3. Applications

**Test-time training.** Here, we use our PRM to train reasoning policies on new datasets to evaluate the generalizability of our PRM. Specifically, we fix the PRM (*i.e.,* fine-tuned from *Qwen3-4B-Base* on our training set) and use it to train a new *Qwen3-4B-Base* policy model from scratch on the MATH dataset (Hendrycks et al., 2021). Crucially, no outcome-level rewards are provided in this setting; the policy is trained solely under the guidance of the PRM. We evaluate the resulting policy on Math-500, AIME-2024, and OlympiadBench, comparing it against RLOO trained with outcome rewards. Figure 1a shows that our PRM serves as an effective reward model, substantially improving the performance of the base policy. It also achieves a comparable performance with the policy trained from verifiable outcome reward. This result shows that rePIRL can be used for test-time training when outcome reward is not available.

**Test-time scaling.** We further validate the utility of our PRM in Test-time scaling (TTS). In this setting, the PRM

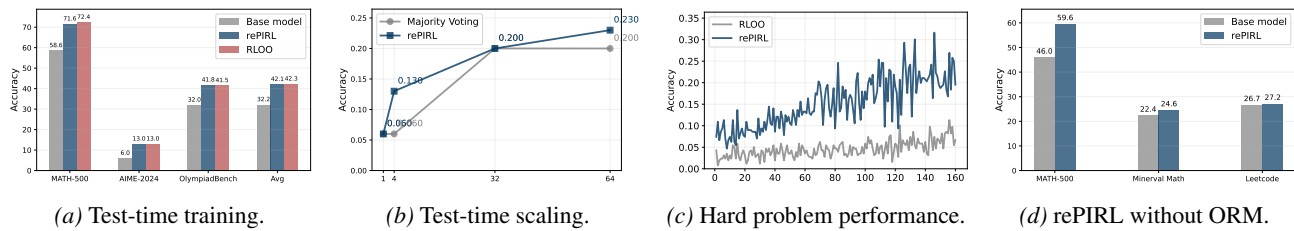

*(a) Test-time training.*     *(b) Test-time scaling.*     *(c) Hard problem performance.*     *(d) rePIRL without ORM.*

*Figure 1.* Performance of three applications of our PRM (Section 4.3) and rePIRL without outcome reward (Section 4.4).

serves as a verifier to score candidate responses generated by the *Qwen3-4B-Base* policy. We define the trajectory score as the mean token-level reward derived from the PRM. We benchmark this approach against a standard majority voting baseline in AIME-2024. Figure 1b shows that employing our PRM as the selector yields monotonic performance improvements as the sampling budget increases, consistently outperforming majority voting.

**PRM efficiency on hard problems.** Here, we demonstrate that our PRM provides informative reward signals early in training, thereby accelerating policy optimization. Specifically, we construct a difficult training subset using problems from the MATH benchmark (Hendrycks et al., 2021) that the base *Qwen3-4B-Base* model fails to solve, and train the policy on this subset using either our PRM or a verifiable outcome reward. As shown in Figure 1c, training with the outcome reward (RLOO) results in consistently low accuracy during the early stages and exhibits little change over a prolonged period, indicating that outcome rewards provide limited useful feedback early on. In contrast, training with our PRM (rePIRL) yields measurable improvements early on, converges substantially faster, and achieves higher accuracy on hard problems. This highlights the utility of PRM when outcome rewards are sparse or insufficient.

### 4.4. rePIRL with PRM only

In the main experiments, the reward function combines stepwise PRM scores with a binary outcome reward; here, we remove the outcome component entirely. As a result, the policy is optimized only to generate reasoning trajectories that the PRM rates highly, without receiving any direct feedback on final correctness. This setup enables us to assess how effectively process-based rewards alone can guide the model toward correct solutions.

We evaluate this variant on Math-500, MinervaMath, and Leetcode based on *Qwen2.5-3B-Instruct*. The results in Figure 1d show that training exclusively with process-level rewards yields substantial gains over the Qwen base model, including a 13.6% improvement on Math-500. These findings suggest that our IRL framework effectively recovers the latent reward function consistent with the expert policy.

Training recipe is important for any post-training reasoning methods. In this paper, we also conduct a rigorous ablation study to explore an effective training recipe for our method, including base model selection, expert data selection, and RL training algorithm. The details are shown in Appendix D. We believe our findings and experimental methods can benefit future works in this domain.

## 5. Discussion

**Training efficiency.** We emphasize that rePIRL does not introduce significant computational overhead compared to many state-of-the-art methods. By leveraging RLOO for non-parametric value estimation, rePIRL eliminates the need for a value network, restricting optimization to only the policy and reward models. This architectural complexity aligns with methods requiring PRMs, such as PRIME and Math-Shepherd. For instance, when fine-tuning *Qwen2.5-3B-Instruct* on math tasks, rePIRL requires approximately 13.3 training hours compared to 14.6 hours for PRIME. Relative to outcome-based RL baselines like RLOO, rePIRL introduces only a $\sim$1.4$\times$ training time overhead (*e.g.,* 13.3 hours vs. 9 hours to fine-tune *Qwen2.5-3B-Instruct* on math tasks). Importantly, this limited overhead is offset by several key advantages. Beyond efficiency, rePIRL provides a unified framework; as detailed in Section 3.3. Empirically, this formulation results in consistent performance gains over all baselines across diverse tasks and model architectures. Moreover, the learned PRM serves as a reusable and standalone module, benefiting a number of downstream applications such as test-time training and scaling at Section 4.3.

**Training stability and dynamics.** Adversarial training is well known to suffer from instability, a pattern we also observed in our preliminary experiments. To mitigate this, we enhance reward model training with corrected trajectories from the current policy, treating them as additional expert data while retaining only failed trajectories as policy data (Algorithm 1). This signals to the reward model that these corrected trajectories should receive high reward, yielding more reliable gradients for policy updates. As shown in our ablation study in Appendix D, this component substantially improves both training stability and final performance.

Furthermore, our analysis of training dynamics (Appendix E) confirms that rePIRL achieves accelerated convergence and superior stability relative to established baselines. To further characterize this stability, we analyzed the evolution of policy entropy using the *Qwen3-4B-Base* model on mathematical reasoning tasks. While the base model exhibits an initial entropy of ∼0.80, rePIRL and RLOO retain relatively higher entropy levels after two epochs (0.11 and 0.12, respectively). Conversely, the Intuitor (Zhao et al., 2025b) baseline exhibits a severe entropy collapse to 0.028, indicative of excessive mode-seeking behavior. Although policy entropy naturally decays during fine-tuning, our empirical results indicate that rePIRL prevents over-determinism and matches the regularization capabilities of standard RL baselines.

## 6. Conclusion and Future Works

In this paper, we presented rePIRL, an IRL–inspired framework for learning PRMs with minimal assumptions. We introduce a dual learning process that alternates between policy and PRM updates, along with techniques tailored to scaling inverse RL for LLMs. We prove that rePIRL can unify online and offline PRM learning under additional assumptions. Our extensive experiments on math and coding reasoning benchmarks demonstrate the effectiveness of our methods, the utility of our PRM, and validate our training recipe.

Our work points to a few interesting future work directions. First, it would be valuable to systematically study and compare the various heuristics proposed for online RL with outcome reward for LLM reasoning (*e.g.,* curriculum learning (Bengio et al., 2009), self-correction (Shinn et al., 2023)), comparing their effectiveness under our learning framework. Second, rePIRL could be combined with complementary approaches for learning PRMs, *e.g.,* the intrinsic reward methods. It is interesting to investigate the proper reward shaping strategies when comparing rePIRL with other PRMs as well as different outcome rewards. Finally, investigating scaling rePIRL to agentic tasks and larger models may unlock the potential of PRMs for broader application domains and enable stronger performance.

## Acknowledgement

This work was supported in part by ARL Grant W911NF-23-2-0137. We gratefully acknowledge the support of UC Noyce, FAR AI, OpenAI, anthropic, Google, and Berkeley RDI.

## Impact Statement

This paper proposes rePIRL, a framework for improving LLM reasoning through inverse RL, with the goal of enabling efficient training of PRMs without reliance on costly token-level human supervision or MCTS-based methods.

Empirically, rePIRL yields substantial gains in automated code generation (*i.e.,* Leetcode and LiveCodeBench), which can improve developer productivity and accelerate software development. At the same time, enhanced coding capabilities in LLMs present dual-use risks, including the potential to lower the barrier for generating malicious code or exploiting software vulnerabilities. Mitigating these risks requires complementary safeguards such as controlled deployment, monitoring, and responsible use guidelines.

From an environmental perspective, although training LLMs remains computationally intensive, rePIRL is designed to be parameter- and compute-efficient. In particular, it avoids training separate value networks and achieves state-of-the-art performance with training costs comparable to or lower than existing baselines, thereby helping to limit the additional environmental footprint associated with model training.

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

# A. Theoretical Proof

## A.1. Proof of Theorem 1

We consider the entropy-regularized objective at a fixed state $s$ with action-values $Q^*(s, a)$:

$$\max_{\pi(\cdot|s)} \ \mathcal{L}(\pi) = \sum_a \pi(a|s)Q^*(s,a) + \beta\Big[-\sum_a \pi(a|s)\log \pi(a|s)\Big], \qquad \sum_a \pi(a|s) = 1.$$

Introducing a Lagrange multiplier $\lambda$ for the normalization constraint, the Lagrangian is

$$\mathcal{J}(\pi, \lambda) = \sum_a \pi(a)\Big[Q^*(s,a) - \beta \log \pi(a)\Big] + \lambda\Big(1 - \sum_a \pi(a)\Big).$$

Setting the derivative w.r.t. $\pi(a)$ to zero:

$$\frac{\partial \mathcal{J}}{\partial \pi(a)} = Q^*(s,a) - \beta(1 + \log \pi(a)) - \lambda = 0 \quad \Rightarrow \quad \pi(a) \propto \exp\Big(\frac{Q^*(s,a)}{\beta}\Big).$$

Normalizing over $a$ yields the optimal policy:

$$\pi_*(a|s) = \frac{\exp\big(Q^*(s,a)/\beta\big)}{\sum_{a'} \exp\big(Q^*(s,a')/\beta\big)}.$$

Substituting $\pi^*$ into $V^*$ gives the soft value function

$$V^*(s) = \beta \log \sum_{a'} \exp\big(Q^*(s,a')/\beta\big),$$

so that $\pi_*(a|s) = \exp(\frac{1}{\beta}[Q^*(s,a) - V^*(s)])$.

## A.2. Proof of Proposition 1

Starting from the entropy-regularized optimal policy: $\pi_*(a|s) = \exp(\frac{1}{\beta}[Q^*(s,a) - V^*(s)])$

Taking logs and subtracting $\log \pi_{\text{ref}}(a|s)$:

$$\log \frac{\pi_*(a|s)}{\pi_{\text{ref}}(a|s)} = \frac{Q^*(s,a) - V^*(s)}{\beta} - \log \pi_{\text{ref}}(a|s).$$

In the bandit case, $Q^*(s,a) = r'(s,a) = r(s,a) + \beta \log \pi_{\text{ref}}(a|s)$, giving

$$\log \frac{\pi_*(a|s)}{\pi_{\text{ref}}(a|s)} = \frac{r(s,a) - V^*(s)}{\beta}.$$

Exponentiating both sides leads to

$$\pi_*(a \mid s) = \frac{1}{Z_*(s)} \pi_{\text{ref}}(a \mid s) \exp\Big(\frac{1}{\beta} r(a, s)\Big), \quad Z_*(s) = \exp\Big(\frac{V^*(s)}{\beta}\Big).$$

## A.3. Proof of Proposition 2

In DQO, the authors adopt the Soft Actor-Critic (SAC) framework to learn the state-value function $V$ and state-action value function $Q$. In SAC, these functions are optimized by minimizing the squared Bellman residuals:

$$L_V(\phi) = \mathbb{E}_{(s_t, a_t, s_{t+1}) \sim \mathcal{D}}\Big[\big(V_\phi(s_t) - Q_\theta(s_t, a_t) + \beta \log \pi_\theta(a_t \mid s_t)\big)^2\Big], \tag{a.3.1}$$

$$L_Q(\theta) = \mathbb{E}_{(s_t, a_t, s_{t+1}) \sim \mathcal{D}}\Big[\big(Q_\theta(s_t, a_t) - r(s_t, a_t) - V_\phi(s_{t+1})\big)^2\Big]. \tag{a.3.2}$$

At convergence, these updates satisfy the soft Bellman equations:

$$\hat{V}^\pi(s_t) = \mathbb{E}_{a_t \sim \pi(\cdot \mid s_t)}\Big[\hat{Q}^\pi(s_t, a_t) - \beta \log \pi(a_t \mid s_t)\Big], \quad \hat{Q}^\pi(s_t, a_t) = r(s_t, a_t) + \gamma \, \mathbb{E}_{s_{t+1}}\big[\hat{V}^\pi(s_{t+1})\big].$$

which correspond to the soft maximum-entropy RL in Theorem 1.

Motivated by DPO, DQO further reparameterizes the $Q$-function directly in terms of the policy:

$$Q_\theta(s_t, a_t) = V_\phi(s_t) + \beta \log \pi_\theta(a_t \mid s_t), \tag{a.3.3}$$

where $\pi_\theta$ denotes the policy network. Notably, (a.3.3) coincides with the optimal policy $\pi^*$ under the soft maximum-entropy RL formulation, as established in Theorem 1. By substituting (a.3.3) into (a.3.2) and rewriting (a.3.1) by replacing the $Q$-function using (a.3.2) accordingly, DQO eliminates the explicit dependence on the $Q$-function. Taken together, DQO optimizes the same maximum-entropy RL objective for the policy model as our method.

## A.4. Proof of Proposition 3

Our main idea is to show that the gradient of Eqn. (5) is equivalent to the gradient of $\mathbb{E}_{\tau \sim \mathcal{D}}\big[\log D(\tau)\big] + \mathbb{E}_{\tau \sim \pi_\theta}\big[\log\big(1 - D(\tau)\big)\big]$, where $D = \frac{\tilde{p}}{\tilde{p} + \pi_\theta}$ and $\tilde{p} = \frac{1}{z}\exp(r_\phi(\cdot))$, thereby establishing the equivalence between the two objective functions.

To begin with, the gradient of Eqn. (5) can be written as

$$\begin{aligned}
\partial_\phi \mathcal{L}_{\text{reward}}(\phi) &= \mathbb{E}_{\tau \sim \mathcal{D}}[\partial_\phi r_\phi(\tau)] - \partial_\phi \log\left(\mathbb{E}_{\tau \sim \mu}\left[\frac{\exp(r_\phi(\tau))}{\tilde{\mu}(\tau)}\right]\right) \\
&= \mathbb{E}_{\tau \sim \mathcal{D}}[\partial_\phi r_\phi(\tau)] - \frac{\mathbb{E}_{\tau \sim \mu}\left[\frac{\exp(r_\phi(\tau))}{\tilde{\mu}(\tau)}\partial_\phi r_\phi(\tau)\right]}{\mathbb{E}_{\tau \sim \mu}\left[\frac{\exp(r_\phi(\tau))}{\tilde{\mu}(\tau)}\right]} \\
&= \mathbb{E}_{\tau \sim \mathcal{D}}[\partial_\phi r_\phi(\tau)] - \mathbb{E}_{\tau \sim \mu}\left[\frac{\frac{1}{z}\exp(r_\phi(\tau))\partial_\phi r_\phi(\tau)}{\tilde{\mu}(\tau)}\right].
\end{aligned}$$

Here, $\mu$ denotes a mixture distribution of $\pi_\theta$ and $D$, where $\tilde{\mu}(\tau) = \frac{1}{2Z}\exp\big(r_\phi(\tau)\big) + \frac{1}{2}\pi_\theta(\tau)$.

Note that, $\mathbb{E}_{\tau \sim \mathcal{D}}\big[\log D(\tau)\big] + \mathbb{E}_{\tau \sim \pi_\theta}\big[\log\big(1 - D(\tau)\big)\big]$ with $D = \frac{\tilde{p}}{\tilde{p} + \pi_\theta}$ and $\tilde{p} = \frac{1}{z}\exp(r_\phi(\cdot))$ can be written as

$$\begin{aligned}
\mathcal{L}_{\text{CE}}(D_\phi) &= \mathbb{E}_{\tau \sim \mathcal{D}}\big[\log D(\tau)\big] + \mathbb{E}_{\tau \sim \pi_\theta}\big[\log(1 - D(\tau))\big] \\
&= \mathbb{E}_{\tau \sim \mathcal{D}}\left[\log \frac{\frac{1}{z}\exp(r_\phi(\tau))}{\frac{1}{z}\exp(r_\phi(\tau)) + \pi_\theta(\tau)}\right] + \mathbb{E}_{\tau \sim \pi_\theta}\left[\log \frac{\pi_\theta(\tau)}{\frac{1}{z}\exp(r_\phi(\tau)) + \pi_\theta(\tau)}\right] \\
&= \mathbb{E}_{\tau \sim \mathcal{D}}\left[\log \frac{\frac{1}{z}\exp(r_\phi(\tau))}{\tilde{\mu}(\tau)}\right] + \mathbb{E}_{\tau \sim \pi_\theta}\left[\log \frac{\pi_\theta(\tau)}{\tilde{\mu}(\tau)}\right] \\
&= -\log z + \mathbb{E}_{\tau \sim \mathcal{D}}[r_\phi(\tau)] - \mathbb{E}_{\tau \sim \mathcal{D}}[\log \tilde{\mu}(\tau)] + \mathbb{E}_{\tau \sim \pi_\theta}[\log \pi_\theta(\tau)] - \mathbb{E}_{\tau \sim \pi_\theta}[\log \tilde{\mu}(\tau)] \\
&= -\log z + \mathbb{E}_{\tau \sim \mathcal{D}}[r_\phi(\tau)] + \mathbb{E}_{\tau \sim \pi_\theta}[\log \pi_\theta(\tau)] - 2\mathbb{E}_{\tau \sim \mu}[\log \tilde{\mu}(\tau)].
\end{aligned}$$

Differentiating these terms yields

$$\partial_\phi \mathcal{L}_{\text{CE}}(D_\phi) = \mathbb{E}_{\tau \sim d}[\partial_\phi r_\phi(\tau)] - \mathbb{E}_{\tau \sim \mu}\left[\frac{\frac{1}{Z}\exp(r_\phi(\tau))\partial_\phi r_\phi(\tau)}{\tilde{\mu}(\tau)}\right] = \partial_\phi \mathcal{L}_{\text{reward}}(\phi).$$

which completes the proof.

## A.5. Proof of Proposition 4

We begin with the original proof that derives the intermediate reward in PRIME and outline the assumptions underlying its formulation.

| config | value |
|---|---|
| optimizer | AdamW |
| policy model learning rate | 5e-7 |
| reward model learning rate | 3e-8 |
| weight decay | 0.0 |
| policy model batch size | 128 |
| reward model batch size | 128 |
| policy model mini batch size | 32 |
| reward model mini batch size | 32 |
| policy model micro batch size | 1 |
| reward model micro batch size | 1 |
| max prompt length | 1535 |
| max response length | 3000 |
| reward model gradient clip value | 10.0 |
| policy model gradient clip value | 1.0 |
| clip ratio | 0.2 |
| policy model epoch | 1 |
| reward model epoch | 1 |
| total training epochs | 2 |
| coefficient of entropy policy loss | 0.001 |

*Table 3.* Training hyperparameters for both coding and math dataset.

Specifically, PRIME assumes that the final reward can be expressed as $r = \log \frac{\pi_\theta}{\pi_{\pi_{\text{ref}}}}$ and derives the intermediate reward $r_t$ from the difference in the value function between consecutive time steps:

$$\sum_{i=1}^{t} \beta \log \frac{\pi_\theta(y_i|\mathbf{y}_{<i})}{\pi_{\pi_{\text{ref}}}(y_i|\mathbf{y}_{<i})} - \sum_{i=1}^{t-1} \beta \log \frac{\pi_\theta(y_i|\mathbf{y}_{<i})}{\pi_{\pi_{\text{ref}}}(y_i|\mathbf{y}_{<i})}.$$

Taken together, the ORM is defined over the entire sequence (the outcome), indicating that the outcome reward corresponds to the total reward of the episode. This final reward is perfectly decomposable and can be expressed as a sum of per-step rewards, as shown by $V(s_t) - V(s_{t-1}) = r(s_t, a_t)$.

## B. Training and Inference Details

**Training Details.** For training both our methods and the baselines, we set the maximum prompt length to 1,535 tokens and the maximum response length to 3,000 tokens. Following PRIME, we filter out prompts for which the accuracy exceeds 0.8 or falls below 0.2. For the clipping hyperparameter, we use the default clip ratio of 0.2 for both lower and upper bounds. The policy model uses a gradient clip value of 1.0, while the reward model uses a value of 10.0. The complete set of training hyper-parameters is detailed in Table 3. In the application experiments (Section 4.3), when training the policy from scratch using only our PRM, we introduce an additional format-based reward to mitigate reward hacking. Specifically, the format reward is set to 0 if the model output matches the pattern `.*\\boxed{.*}.*`, and to $-1$ otherwise.

**Inference Parameters.** For inference, we adopt a greedy decoding strategy (*i.e.,* setting the temperature to 0, `top_k` to 1, and `top_p` to 1) for all models. Specifically, we perform inference using the latest version of the VLLM (Kwon et al., 2023) framework to reduce memory usage and accelerate computation. For the testing-time scaling experiments in Section 4.3, we set the temperature to 0.8 while keeping all other hyper-parameters unchanged.

## C. Non-greedy Pass@1 evaluation

While our main results relys on greedy decoding, we additionally evaluate model performance using non-greedy sampling (*i.e.,* set the temperature to 0.6). Specifically, we sample $k = 16$ independent responses per prompt and compute the average pass rate to obtain an unbiased estimate of the expected pass@1, thereby accounting for variance in reasoning paths.

*Table 4.* **Expected pass@1 performance on mathematical benchmarks using stochastic sampling ($T = 0.6$, $k = 16$).** Average pass rates are reported across 16 independent responses per question, accounting for variance in reasoning paths.

| Model | MATH500 | AIME2024 | MinervaMath | AMC | OlympiadBench | Avg. |
|---|---|---|---|---|---|---|
| Qwen2.5-3B-Instruct | 41.7 | 4.7 | 20.7 | 27.5 | 26.6 | 24.2 |
| RLOO | 59.5 | 6.0 | 24.7 | 32.0 | 28.6 | 30.1 |
| PRIME | 54.1 | 7.9 | 23.9 | 31.9 | 27.8 | 29.1 |
| **rePIRL** | 58.9 | 7.5 | 26.6 | 33.2 | 28.2 | **30.8** |
| Qwen3-4B-Base | 49.5 | 5.8 | 15.6 | 27.1 | 26.8 | 24.9 |
| RLOO | 68.6 | 10.2 | 27.0 | 42.7 | 38.4 | 37.3 |
| PRIME | 66.7 | 10.8 | 32.0 | 41.4 | 36.9 | 37.5 |
| **rePIRL** | 69.7 | 12.1 | 29.0 | 43.2 | 39.0 | **38.5** |

*Table 5.* **Performance comparison of different RL algorithms on mathematical and coding benchmarks.** Across all tasks, RLOO with our PRM achieves the best average performance.

| Model | Mathematical Reasoning | | | | | | Coding Reasoning | | |
|---|---|---|---|---|---|---|---|---|---|
|  | MATH500 | AIME2024 | MinervaMath | AMC | OlympiadBench | Avg. | Leetcode | LiveCodeBench | Avg. |
| Qwen2.5-3B-Instruct | 46.0 | 10.0 | 22.4 | 34.9 | 28.9 | 28.4 | 26.7 | 18.3 | 22.5 |
| PPO | 51.4 | 3.3 | 25.0 | 31.3 | 28.3 | 27.9 | 27.2 | 18.9 | 23.0 |
| PPO w/ rePIRL | 56.6 | 13.3 | 25.0 | 28.9 | 27.9 | 30.3 | 33.5 | 20.1 | 26.8 |
| GRPO | 61.6 | 3.3 | 26.1 | 36.1 | 28.6 | 31.1 | 29.0 | 19.2 | 24.1 |
| GRPO w/ rePIRL | 62.2 | 10.0 | 27.4 | 36.5 | 29.1 | 33.0 | 34.3 | 20.1 | 27.2 |
| RLOO | 60.1 | 3.3 | 25.7 | 35.2 | 29.1 | 31.7 | 28.9 | 19.4 | 24.1 |
| RLOO w/ rePIRL | 62.4 | 10.0 | 27.2 | 38.5 | 29.3 | **33.5** | 35.0 | 20.3 | **27.7** |

Under this setting (*i.e.,* temperature = 0.6, $k = 16$ samples per prompt), we compare rePIRL against two strong baselines (*i.e.,* RLOO and PRIME) on our math benchmarks. The results are summarized in Table 4.

While the absolute performance numbers shift as expected under non-greedy decoding, our relative performance gains remain significant and consistent. In particular, rePIRL continues to outperform the strongest baselines , achieving average improvements of $0.7\%$ over RLOO on *Qwen2.5-3B-Instruct* and $1.0\%$ over PRIME on *Qwen3-4B-Base*.

## D. Ablation study

Unless otherwise specified, all ablation study experiments are conducted using the *Qwen2.5-3B-Instruct* model.

### D.1. Model and Data Selection

Here, we study two key factors: the teacher model for expert trajectories and the structure of the reward model.

**Replacing Claude with open-source models.** We further evaluate rePIRL by replacing the Claude-3.7-Sonnet model (already not the latest Claude model) with the open-source DeepSeek-R1-Reasoning model (Guo et al., 2025) for collecting expert trajectories. The performance on math tasks is shown in Figure 2. The comparable results using Claude and Deepseek-R1-Reasoning as expert models further demonstrate that rePIRL does not rely on costly proprietary models to generate expert trajectories.

**Varying the reward model architecture.** We conducted additional ablation studies on the reward model's architecture. The current rePIRL results utilize a PRM initialized from the same base language model as the policy model. We systematically varied the reward model sizes and architectures as shown in Table 6. From this table, we observe that replacing the reward model with a smaller one degrades performance. Nevertheless, our approach still outperforms the RLOO baselines, demonstrating that rePIRL generalizes across different reward model architectures and sizes. We note that using Qwen models for experiments and ablation is standard practice, as the Qwen family represents the best-performing open-source model series. Different Qwen models have distinct architectural properties that enable controlled comparisons. We do not consider other LLM families (*e.g.,* LLaMA) or non-LLM architectures, given the performance differences in their base

*Table 6.* **Performance comparison across different reward model configurations on math tasks**. Here, P-3B-R-1.5B denotes that the policy model is *Qwen2.5-3B-Instruct* and reward model is *Qwen2.5-1.5B-Instruct*.

| Model | MATH-500 | AIME-2024 | MinervaMath | AMC | Olympiadbench | Avg. |
|---|---|---|---|---|---|---|
| P-3B-R-3B | 62.4 | 10.0 | 27.2 | 38.5 | 29.3 | 33.5 |
| P-3B-R-1.5B | 57.8 | 10.0 | 27.6 | 36.1 | 27.6 | 31.8 |
| P-4B-R-4B | 72.8 | 16.0 | 33.5 | 43.3 | 42.5 | 41.6 |
| P-4B-R-1.7B | 72.6 | 10.0 | 34.9 | 37.3 | 38.7 | 38.7 |

*Table 7.* **Performance comparison across different rollout strategies on math tasks**. Here, rePIRL w/o correct traj from policy indicates that correct trajectories produced by the policy are not included as pseudo-expert trajectories. We utilize the *Qwen2.5-3B-Instruct* as our base model.

| Model | MATH-500 | AIME-2024 | MinervaMath | AMC | Olympiadbench | Avg. |
|---|---|---|---|---|---|---|
| rePIRL | 62.4 | 10.0 | 27.2 | 38.5 | 29.3 | 33.5 |
| rePIRL w/o correct traj from policy | 61.4 | 3.3 | 31.6 | 34.9 | 27.6 | 31.7 |

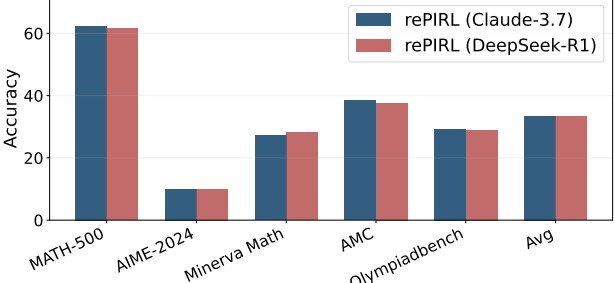

*Figure 2.* Comparison of rePIRL using Claude-3.7-Sonnet versus DeepSeek-R1 as expert trajectory generators.

*Figure 3.* Ablation study comparing rePIRL with importance sampling versus baselines with only increased policy updates.

models.

### D.2. Learning Algorithms

For learning algorithms, we focus on the tricks and design choices highly related to our method, including the choice of policy updating method, reward model and policy training scheduling, and rollout strategies. We do not study the standard tricks, such as clip high, no KL terms.

**Choice of the policy update method.** We evaluate rePIRL with different RL algorithms by modifying only the advantage estimation while keeping the clipped surrogate loss fixed. We implement rePIRL with PPO, GRPO, and RLOO, where PPO learns an additional value function and GRPO normalizes expected total reward. As shown in Table 5, rePIRL consistently improves performance across all algorithms, demonstrating broad applicability. RLOO with process rewards achieves the best overall performance, while PPO shows minimal improvement despite added computational cost from its critic model. These results indicate RLOO provides the most effective advantage estimation for process-based rewards, leading us to adopt it as the default method in subsequent experiments.

**Importance sampling vs update epochs.** Note that we apply importance weighting when updating the reward model in Section 3.2. We additionally compare our method against a baseline that simply increases the number of inner policy updates (*i.e.,* motivated by standard GAN training). Across all math tasks, our method consistently outperforms this heuristic baseline as shown in Figure 3. These results demonstrate that the performance gains stem from the necessity of the importance-sampling correction itself, rather than from merely performing additional policy updates.

**Varying rollout strategy on training stability and performance.** Table 7 studies the impact of including corrected policy trajectories as pseudo-expert data during reward model training. Across all evaluated math benchmarks, rePIRL consistently outperforms the variant that excludes correct trajectories generated by the policy. By explicitly labeling corrected trajectories as expert data while retaining only failed trajectories as policy samples, the reward model learns a more reliable reward

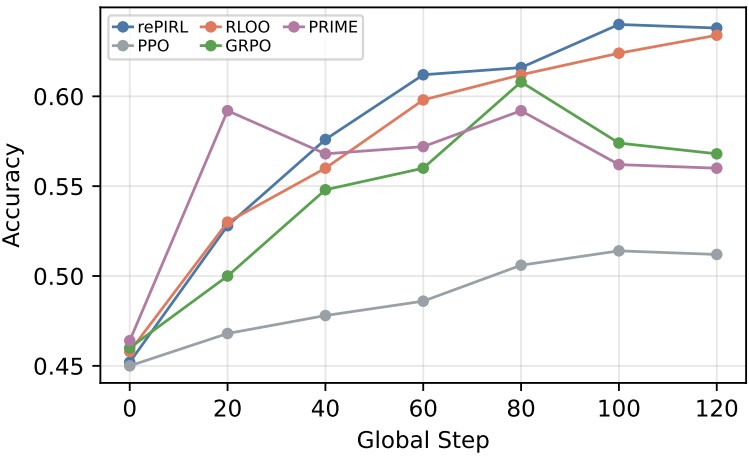

*Figure 4.* Comparison of training dynamics on MATH-500 validation set.

landscape, yielding more stable optimization and improved final performance.

## E. Training dynamics

We illustrate the training dynamics of PRIME, RLOO, PPO, GRPO and rePIRL in Figure 4. Following (Zha et al., 2025), we use MATH-500 as a validation set to compare the learning curves of these algorithms, with experiments conducted on math datasets using the Qwen2.5-3B-Instruct model. Overall, rePIRL converges faster and exhibits the most stable performance among all evaluated methods.

