# OpenReview forum: "rePIRL: Learn PRM with Inverse RL for LLM Reasoning"
_ICML.cc/2026/Conference — ICML 2026 regular_

### Official Review · Reviewer_gP2S · 2026-03-10

**Soundness:** 2
**Presentation:** 2
**Significance:** 3
**Originality:** 3
**Overall Recommendation:** 4
**Confidence:** 3

**Summary:**

This paper proposes rePIRL, a framework for learning Process Reward Models (PRMs) for LLM reasoning by adapting classical Inverse Reinforcement Learning (IRL). The core idea is to recover a reward function from expert trajectories alone, without token-level annotations, the expert policy itself, or its reward function, and then co-train a policy using maximum entropy RL.

**Compliance With Llm Reviewing Policy:**

Affirmed.

**Final Justification:**

Overall, most of my previous concerns have been addressed, though I still have a few remaining questions regarding the comparisons with PRIME. I appreciate the additional experiments and the promising empirical results presented in this revision, and I am therefore increasing my rating to 4.

**Key Questions For Authors:**

- Can the authors provide training curves for rePIRL vs. PRIME / RLOO / PPO / GRPO? The paper emphasizes that rePIRL jointly updates the PRM and policy and claims empirical gains, but training curves would be very helpful to assess convergence speed, optimization stability, and sample efficiency, especially since the method relies on alternating reward/policy learning and importance weighting.
- How stable is the importance weighting term in practice? In general I will assume the importance weight has high variance.
- To what extent do the gains over PRIME come from the proposed algorithm versus the additional expert demonstrations used by rePIRL?

**Limitations:**

See the weakness above.

**Strengths And Weaknesses:**

Strengths:
- Generalizing IRL to token-level LLM reasoning is a genuinely interesting idea. The MDP formulation is clean, and the derivation from energy-based modeling through importance sampling to a tractable algorithm is well-motivated. The framing around Table 1 is compelling.
- The experiments and ablation study are comprehensive. Testing across seven benchmarks spanning both math and coding, with two model families and multiple baselines, provides convincing evidence of generalizability.

Weakness:
- I feel the importance sampling weights may be poorly behaved. The weight wj involves the product of token-level probabilities in the denominator, which is related to the trajectory lenth. For responses of more than 1k tokens, this leads to severe high-variance. The authors did not analyze the effective sample size of these weights. How would authors tackle this issue?
- Entropy collapse claim needs stronger support. The paper claims rePIRL avoids entropy collapse (Table 1), but provides no direct empirical measurement of policy entropy over training.
- On Algorithm 1: Step 7 applies importance weights exclusively to rollouts where r(\tau) = 0. This asymmetric treatment is not derived from the theoretical framework in Equation 5, which makes no distinction between correct and incorrect policy rollouts when computing importance-weighted negative gradients. The authors should provide stronger justification for this design choice. In particular, if successful policy rollouts were instead retained in the importance-weighted negative term rather than being reclassified as expert data, how would the training dynamics and final performance change?
- The comparison with PRIME is still not entirely convincing. rePIRL benefits from expert demonstrations generated by Claude-3.7, while PRIME is designed as an online method without access to such expert supervision. Therefore, the two methods are not compared under fully matched supervision budgets, and the current results do not cleanly isolate the advantage of the proposed algorithm itself. Providing training curves would further strengthen this comparison by revealing convergence behavior, stability, and sample efficiency. In addition, Table 4 is difficult to interpret. The fact that GRPO/PPO trained on AIME2024 does not appear to improve performance on AIME2024 itself is surprising, and the current presentation risks being misleading without further explanation. The authors should clarify the experimental takeaway and discuss why in-domain training does not translate into clearer in-domain improvement.

---

> ### Author Rebuttal · Authors · 2026-03-30
>
> > The sampling weights may be poorly behaved and have large variance
>
> **Response:**
> We thank the reviewer for highlighting this critical point regarding the stability of importance sampling in long-horizon tasks. We address the potential for high variance and numerical instability through the following implementation details:
> * **Numerical Stability via Log-Space Computation**: To prevent numerical overflow or underflow (the "explode" issue), we do not compute $w$ directly, Instead, we perform all calculations in log-space: $\log w = r_\phi(\tau) - \sum_{t=1}^{|\tau|} \log \pi_\theta(a_t | s_t)$.  The final normalized weights are then derived using a **Softmax transformation** across the mini-batch (size 32), which incorporates the log-sum-exp trick. This ensures that even for sequences exceeding 1k tokens, the weights remain numerically well-behaved.
>
> * **Variance Reduction via  Sample Filtering**: Following the protocol in PRIME, we filter out prompts for which the accuracy is extreme (exceeding 0.8 or falling below 0.2). By focusing on samples with higher high-entropy outcomes, we ensure that the importance weights $w$ remain within a manageable range, preventing a few samples from dominating the gradient.
>
> * **Empirical Stability and Diagnostics**: In practice, the combination of log-space normalization, sample filtering, and a mini-batch size of 32 provides a consistent training signal for the PRM. To quantify this stability, we monitored the **Effective Sample Size (ESS)** throughout training. We observed an average ESS of approximately 6 per mini-batch of 32, which indicates sufficient particle diversity to provide a reliable gradient estimate without collapsing.
>
> * **Comparison with Heuristic Baselines**: To further justify our approach, we compared our IS method against a baseline that simply increases the number of inner policy updates (akin to standard GAN training). As shown in **Figure 3 (Appendix C.2)**, our principled importance sampling consistently outperforms this heuristic baseline across all math tasks, confirming that the IS signal is both stable and informative.
>
> > Entropy collapse claim needs stronger support.
>
> **Response:**
>
> We appreciate the reviewer’s suggestion to provide stronger empirical support for our claim regarding entropy collapse. To address this, we conducted a comparative analysis of entropy changes before and after training across different methods using the Qwen3-4B-Base model on math datasets.
>
> The initial entropy of Qwen3-4B-Base is ~0.80 on the math training set. After training for 2 epochs, we observed the following convergence behaviors:
> * **RePIRL**: Maintained an entropy of 0.11, demonstrating high stability and a healthy distribution for exploration.
> * **RLOO**: Achieved an entropy of 0.12, matching the robustness of established RL baselines.
> * **Intuitor**: Suffered a significant collapse, with entropy dropping to 0.028.
>
> While all fine-tuning methods naturally reduce entropy as the model learns specific patterns, rePIRL avoids the extreme "mode-seeking" behavior seen in Intuitor. By maintaining an entropy level equal to RLOO, RePIRL ensures the model doesn't become overly deterministic.
>
> > Training dynamics.
>
> **Response:**
> We illustrate the training dynamics of PRIME, RLOO, PPO, GRPO and rePIRL in Figure https://tinyurl.com/myvu4ww7. Following [1], we use MATH-500 as a validation set to compare the learning curves of these algorithms, with experiments conducted on math datasets using the Qwen2.5-3B-Instruct model. Overall, RePIRL converges faster and exhibits the most stable performance among all evaluated methods.
>
>
> [1] RL Tango: Reinforcing Generator and Verifier Together for Language Reasoning
>
> > How much of RePIRL’s gain over PRIME comes from the algorithm vs. added expert demonstrations?
>
>
> **Response:**
> Please refer to our response to Reviewer x8Gu regarding the same concern: “The method requires a batch of expert demonstrations for the entire training dataset, whereas PRIME does not.”
>
> > How would retaining successful rollouts in the negative term (instead of relabeling them as expert data) affect training dynamics and performance?
>
> **Response:**
> We relabel correct trajectories as expert data to improve training stability. In our preliminary experiments, we observed a significant imbalance early in training: the PRM quickly became overly strong, leaving the relatively weak policy with limited and uninformative learning signals. Inspired by training dynamics in GANs, we address this issue by augmenting reward model training with corrected trajectories from the current policy (treating them as additional expert data), while retaining only failed trajectories for policy optimization.
>
> As shown in our ablation study (**Table 6 in Appendix C.2**), rePIRL consistently outperforms the variant that does not reclassify correct policy-generated trajectories as expert data across all evaluated math benchmarks, validating the effectiveness of our design choice.

---

> > ### Author Rebuttal · Reviewer_gP2S · 2026-04-03
> >
> > I appreciate the additional experiments and clarifications provided by the authors. Most of my previous concerns have been addressed, though I still have a few questions regarding the comparisons with PRIME (see comments below). That said, given the strong empirical performance and the overall clarity of the presentation, I am happy to increase my rating.

---

### Official Review · Reviewer_x8Gu · 2026-03-11

**Soundness:** 2
**Presentation:** 3
**Significance:** 3
**Originality:** 3
**Overall Recommendation:** 4
**Confidence:** 4

**Summary:**

The authors propose an algorithm to learn a PRM by inverse RL on expert demonstrations. It alternates between updating the PRM by maximizing likelihood of expert trajectories relative to failed policy rollouts, and updating the policy by RLOO where the the outcome reward is combined with a small bonus from the PRM.

**Compliance With Llm Reviewing Policy:**

Affirmed.

**Final Justification:**

The rebuttal provided a more complete context and reinforced my prior assessment.

**Key Questions For Authors:**

1- The importance sampling ratio $w$ is exponential in sequence length in both numerator and denominator. Can you report effective sample size or max-weight fraction during training? This would clarify whether self-normalization is enough or a few weights dominate the reward update.

2- Can you provide multi-seed variance estimates for (at least some of) the main results?

**Limitations:**

yes

**Strengths And Weaknesses:**

Strengths:
- They extract a reward function from a dataset of expert trajectories. This would be useful because the reward function is then composable with other reward functions, providing more options for fine-tuning than a dataset.
- Showing thet DPO, DQO, PRIME and Math-Shepherd are all special cases under additional assumptions. This clarifies this cluttered landscape.

Weaknesses:
- Table 2 can be more rigorous. The numbers are very close and there is no reported std. For example, AIME has 30 questions, and the shown improvement is about 1-2 questions which are often well within std of multiple seeds. MCTS (the strongest baseline) is not fair. Math-Shepherd PRM is trained on GSM8k/MATH data (highly saturated, little signal), but rePIRL PRM is trained on competition-level problems.
- Table 1 and the motivation don't mention a very important downside of rePIRL: The method needs a batch of expert demonstrations for the whole training dataset. PRIME doesn't need this.

---

> ### Author Rebuttal · Authors · 2026-03-30
>
> > Provide the multi-seed evaluation results.
>
> **Response:**
> Thank you for this suggestion. To ensure the robustness of our results, we provide an evaluation that exceeds standard multi-seed reporting by calculating the **expected pass@1** [1]  over $n=16$ samples per problem.
>
> While our initial results used greedy decoding, we have now conducted a more rigorous assessment using non-greedy sampling (i.e., set the temperature to 0.6). By averaging the individual pass rates across 16 independent responses per question, we obtain an unbiased estimator of the model's performance that accounts for variance in reasoning paths.
>
> This approach provides a more comprehensive view of the model's true reasoning capabilities than a few fixed-seed greedy runs. Using this protocol (temperature = 0.6, k=16 samples per prompt), we further compare rePIRL against two strong baselines (RLOO and PRIME) and report the expected pass@1 on our math benchmarks. The results are summarized below:
>
> | Qwen2.5-3B-Instruct | MATH500 | AIME 2024 | MinervaMath | AMC | OlympiadBench | Average |
> |---|---|---|---|---|---|---|
>  | Base | 0.417 | 0.047 | 0.207 | 0.275 | 0.266 | 0.242 |
>  | RLOO | 0.595 | 0.060 | 0.247 | 0.320 | 0.286 | 0.301 |
>  | PRIME | 0.541 | 0.079 | 0.239 | 0.319 | 0.278 | 0.291 |
> | rePIRL | **0.589** | **0.075** | **0.266** | **0.332** | **0.282** | **0.308** |
>
> | Qwen3-4B-Base | MATH500 | AIME 2024 | MinervaMath | AMC | OlympiadBench | Average |
> |---|---|---|---|---|---|---|
> | Base | 0.495 | 0.058 | 0.156 | 0.271 | 0.268 | 0.249 |
> | RLOO | 0.686 | 0.102 | 0.270 | 0.427 | 0.384 | 0.373 |
> | PRIME | 0.667 | 0.108 | 0.320 | 0.414 | 0.369 | 0.375 |
> | rePIRL | **0.697** | **0.121** | **0.290** | **0.432** | **0.390** | **0.385** |
>
>
> While the absolute performance numbers shift as expected under non-greedy decoding, **our relative performance gains remain significant and consistent**. In particular, rePIRL continues to outperform the strongest baselines , achieving average improvements of 0.7% over RLOO on Qwen2.5-3B-Instruct and 1.0% over PRIME on Qwen3-4B-Base. We will include these non-greedy evaluations in the next version.
>
> [1] Evaluating Large Language Models Trained on Code, OpenAI
>
> > Compare with Math-Shepherd PRM is unfair.
>
> **Response:**
> We appreciate the reviewer raising this point. To address this concern and ensure a rigorous comparison, we have evaluated **rePIRL** against a state-of-the-art MCTS baseline utilizing the highly capable **Qwen2.5-Math-PRM-7B** [1] model.
>
> The experiments were conducted using the Qwen2.5-3B-Instruct in the math dataset. We report results for both greedy decoding and pass@1 (averaged over 16 runs at temperature 0.6):
> As shown in the results https://tinyurl.com/3z6e6m6m, while integrating a state-of-the-art PRM noticeably strengthens the MCTS baseline, it still underperforms our method. Specifically, rePIRL surpasses this strong MCTS baseline by an average absolute margin of 1.5% under greedy decoding and 2.4% under pass@1 evaluation.
>
> [1] https://huggingface.co/Qwen/Qwen2.5-Math-PRM-7B
>
>
> > The sampling ratio is exponential in sequence length, leading to high variance.
>
> **Response:**
> Please refer to our response to Reviewer gP2S regarding the same question: “The sampling weights may be poorly behaved and have large variance.”
>
>
> > The method needs a batch of expert demonstrations for the whole training dataset. PRIME doesn't need this.
>
>
> **Response:** To ensure a fair comparison with PRIME, we introduce a baseline that combines expert demonstrations with PRIME (i.e., SFT + PRIME). In this setting, the pre-trained policy is first warm-started using expert trajectories, followed by applying PRIME on the training dataset. This variant represents a natural approach that leverages both expert demonstrations and PRIME’s online policy updates.
>
> The results are reported in the table below. Interestingly, this hybrid baseline performs worse than using PRIME alone. We hypothesize that SFT on expert trajectories may overly constrain the policy, limiting its exploration during subsequent RL optimization.
> | Qwen2.5-3B-Instruct | AMC | MATH500 | AIME 2024 | MinervaMath | OlympiadBench | Average |
> |---|---|---|---|---|---|---|
> | PRIME | 0.313 | 0.562 | 0.066 | 0.265 | 0.290| 0.299 |
> | SFT+PRIME | 0.253 | 0.552 | 0.10 | 0.195 | 0.193 | 0.258|
> | **rePIRL**|**0.385** |**0.624** |**0.10** |**0.272** |**0.293** |**0.335** |
>
> | Qwen3-4B-Base | AMC | MATH500 | AIME 2024 | MinervaMath | OlympiadBench | Average |
> |---|---|---|---|---|---|---|
> | PRIME | 0.433 | 0.724 | 0.10 | 0.324 | 0.384| 0.393 |
> | SFT+PRIME | 0.386 | 0.652 | 0.06 | 0.239| 0.356 | 0.339 |
> | **rePIRL**| **0.433** |**0.728** |**0.16** |**0.335** |**0.425** |**0.416** |
>
> In contrast, our method effectively incorporates expert trajectories via inverse RL without hindering exploration, leading to more consistent and reliable performance gains.

---

> > ### Author Rebuttal · Reviewer_x8Gu · 2026-04-04
> >
> > I thank the authors for their rebuttal. Most of the concerns have been addressed. Regarding the expert requirement compared to PRIME, I mostly meant that this is sometimes a significant constraint that may limit the usage of rePIRL. For the IS ratio, an effective batch size of 6 for a batch of 32 shows an IS ratio of about 0.2, which indicates a not-ideal approximation in eq. 3. The performance is encouraging despite this.

---

> > > ### Author Response · Authors · 2026-04-05
> > >
> > > Thank the reviewer for considering our rebuttal and for the constructive feedback. Please see below for our responses.
> > >
> > > > expert requirement is sometimes a significant constraint that may limit the usage of rePIRL
> > >
> > > **Response:**
> > > rePIRL is not constrained by the expert requirements for the following reasons.
> > >
> > > * **Effectiveness of Open-Weight Models**: While our primary experiments utilize Claude-3.7-sonnet (a proprietary model), we specifically designed an ablation to test whether proprietary APIs are a strict prerequisite. As shown in Figure 2 (Appendix C.1), rePIRL remains highly effective when using accessible, open-weight models like DeepSeek-R1 to generate the expert demonstrations. This demonstrates that our framework can be fully implemented without relying on restrictive or costly private APIs.
> > >
> > > * **One-Time Cost**: The expert requirement is strictly a one-time, offline cost to train the PRM. Once trained, the PRM is a standalone, reusable module. As detailed in Section 4.3, this learned PRM unlocks significant zero-shot utility across downstream applications, including test-time training and test-time scaling. Therefore, the initial data-gathering constraint is heavily amortized by the model's long-term downstream value.
> > >
> > >
> > > >  IS ratio is about 0.2, which indicates a not-ideal approximation in eq. 3
> > >
> > > **Response:**
> > > While an ESS  ratio of ~0.2 does indicate concentrated weights, we would respectfully clarify that this reflects the intended behavior of Eq. 3 rather than a non-ideal approximation.
> > >
> > > The weighting mechanism is explicitly designed to correct for the distribution shift between the current policy and the latent optimal policy. To achieve a theoretically unbiased estimator, the formula must assign higher weights to the subset of samples that best align with the optimal policy. This mathematically necessitates a lower ESS. As the reviewer kindly noted, our strong empirical performance demonstrates that trading off a higher ESS to maintain an unbiased, concentrated gradient signal is highly effective for optimization in practice.
> > >
> > > We hope these additional discussions fully address your remaining concerns. If our response has satisfactorily resolved your questions, we kindly ask that you consider raising your score. Thank you again for your time.

---

### Official Review · Reviewer_6fKw · 2026-03-12

**Soundness:** 3
**Presentation:** 2
**Significance:** 3
**Originality:** 3
**Overall Recommendation:** 3
**Confidence:** 3

**Summary:**

This paper proposes rePIRL, an inverse-RL-inspired framework for learning process reward models (PRMs) for LLM reasoning from expert trajectories without requiring token-level reward annotations, preference labels, or access to the expert policy itself.

**Compliance With Llm Reviewing Policy:**

Affirmed.

**Key Questions For Authors:**

See above

**Limitations:**

yes

**Strengths And Weaknesses:**

### **Strength**
- The paper contains nontrivial ablations. The appendix explores policy optimization variants, reward-model sizes, trajectory selection strategies, and the role of importance sampling, which is helpful for understanding the recipe.
- Conceptual unification is valuable. The attempt to relate offline and online PRM-learning approaches under a common perspective is useful.

### **Weakness**
- There is a substantial gap between the theoretical objective and the actual training algorithm. The implemented algorithm contains several major modifications that are not cleanly justified by the theory: 1. the algorithm uses average process reward rather than the trajectory-level reward used in the derivation, 2. importance weighting appears to be applied only to a subset of trajectories (e.g., failed rollouts), 3. the reward model is additionally trained using corrected policy trajectories as pseudo-expert data.
- The paper does not directly evaluate the learned PRM itself. The central claim is that rePIRL learns an effective process reward model, but the evidence is almost entirely indirect: the PRM helps RL training and rollout selection. This does not fully establish that the learned reward model is semantically correct or stepwise informative.
- The training setup uses problems sampled from PRIME Eurus-2-RL-Data, with expert trajectories generated by Claude-3.7-Sonnet. Do you try any other experts? And this introduces unnegligble computation overhead.

---

> ### Author Rebuttal · Authors · 2026-03-30
>
> > There is a substantial gap between the theoretical objective and the actual training algorithm.
>
> **Response:**
> We now clarify how our practical implementation aligns with the theoretical objective. The algorithm incorporates several adjustments to improve tractability and efficiency, while preserving its core principles:
> * **Regarding (1) - Average vs. Original Reward**: We use the average process reward rather than the raw sum to strictly prevent reward hacking. Without length normalization, the policy learns to exploit the PRM by generating pathologically long or short responses depending on the sign of the reward. This normalization does not violate our derived theoretical objective; it simply scales the process reward by the response length to stabilize the training.
> * **Regarding (2) and (3) - Training Dynamics**: In our preliminary experiments, we observed an imbalance during the early stages of training: the PRM quickly became too strong, leaving the weaker policy model struggling to find a useful learning signal. Drawing inspiration from GAN training dynamics, we mitigate this by enhancing the reward model training with corrected trajectories from the current policy (treating them as additional expert data), while retaining only the failed trajectories for policy training. As demonstrated in our ablation study (Section 5 and Table 6 in Appendix C.2), these practical bridging steps are critical given that they substantially improve both training stability and the final performance of the model.
>
> > This paper does not directly evaluate the learned PRM itself.
>
> **Response:**
> We thank the reviewer for raising this point. While we do not evaluate the PRM in isolation as a standalone classifier, we extensively validate its learned capabilities and accuracy through three distinct downstream applications (detailed in Section 4.3). In all of these settings, the PRM is kept frozen and acts as a plug-in module:
> * **Test-Time Training**: A policy trained entirely using our PRM's signal achieves performance comparable to a policy trained with ground-truth verifiable outcome rewards on new datasets. This demonstrates that the PRM provides a highly accurate proxy for true mathematical correctness.
> * **Test-Time Scaling**: When utilized as a verifier to score candidate responses, our PRM outperforms a strong majority-voting baseline, proving its effectiveness as a reliable evaluator at inference time.
> * **Efficacy on Hard Problems**: We show that our PRM provides dense, informative reward signals early in the training process. This significantly accelerates policy optimization on difficult tasks, whereas relying purely on sparse outcome rewards results in consistently low accuracy during those same early stages.
>
> Together, these three empirical applications robustly demonstrate the effectiveness and the utility of the learned PRM model.
>
> > Try other experts when generate the expert demonstrations.
>
> **Response:**
> Thanks for pointing this out. As detailed in Appendix C.1, we have conducted an ablation study that changes the teacher model from the proprietary Claude model with an open-weight model (DeepSeek-R1). The performance on math tasks, illustrated in Figure 2 (page 18), shows highly comparable results between the two expert models. This demonstrates that rePIRL does not rely on costly proprietary models to generate high-quality expert trajectories, making it both effective and accessible with open-source alternatives.
>
> > Justify the resulting computational overhead by expert trajectories.
>
> **Response:**
> Generating expert trajectories incurs only a one-time precomputation cost. For our math dataset of 7,000 problems, it took approximately 12 hours to generate four expert demonstrations per problem using Claude-3.7-Sonnet. In contrast, existing PRM-based methods (such as MCTS and behavior cloning) also require expert trajectories and are often significantly more time-consuming; MCTS, for instance, must sample until termination at every step. Therefore, our approach does not introduce additional computational overhead compared to standard PRM baselines. Importantly, this initial upfront cost is offset by two key advantages: first, rePIRL provides a unified framework (Section 3.3) that yields consistent performance gains across diverse tasks and architectures; second, the learned PRM serves as a standalone, reusable plug-in module for downstream applications (Section 4.3).

---

> > ### Author Rebuttal · Reviewer_6fKw · 2026-04-03
> >
> > Thank the authors for their feedback. However, my concerns remain.
> > - You should use popular benchmarks to test the performance of PRMs to prove their effectiveness, which is more direct.
> > - Compared with GRPO using outcome rewards solely, this method introduces an unnegligible computation overhead, which is a tradeoff to be considered.

---

> > > ### Author Response · Authors · 2026-04-05
> > >
> > > Thank the reviewer for considering our rebuttal and for the constructive feedback. Please see below for our responses.
> > >
> > > > use popular benchmarks to test the performance of PRM
> > >
> > > **Response:**
> > >
> > > To further validate the effectiveness of our PRM, we evaluated it on ProcessBench [1], a popular benchmark designed to identify the earliest erroneous step in mathematical reasoning.
> > >
> > > We compare our method with two strong PRM baselines: Math-Shepherd (MCTS) and PRIME. Both RePIRL and PRIME operate at the token level, whereas Math-Shepherd is step-level. For a fair comparison, we adapt Math-Shepherd into a token-level MCTS variant with a token-level score. We define the score of each step using the last token and identifying the step with the minimum score as the earliest error. The results are shown below:
> > >
> > > |PRM |Avg.  F1 |
> > > |---|---|
> > > |Token-level MCTS|0.196|
> > > |PRIME|0.192 |
> > > |rePIRL|**0.206**|
> > >
> > > Overall, our method achieves the highest F1, aligning with its performance in Table 2 and the effectiveness of our PRM across three challenging applications (Section 4.3).
> > >
> > > [1] ProcessBench: Identifying Process Errors in Mathematical Reasoning
> > >
> > > > introduce an unnegligible computation overhead compared with outcome-based methods (e.g., GPRO)
> > >
> > > **Response:**
> > >
> > > While introducing a PRM naturally requires more computation than purely outcome-based methods (e.g., GRPO or RLOO), our analysis demonstrates that this overhead is both modest and well-justified.
> > >
> > > Specifically, compared to a standard outcome-based baseline (RLOO), rePIRL introduces only a ~1.4x increase in training time. For example, fine-tuning Qwen2.5-3B-Instruct on our math tasks takes ~13 hours with rePIRL, compared to 9 hours for the baseline.
> > >
> > > It is important to note that this overhead is a beneficial tradeoff for three key reasons. First, rePIRL provides a **unified framework** (Section 3.3) that consistently improves performance across diverse tasks and model architectures. Second, the learned PRM is **reusable** as a standalone module, supporting downstream applications such as test-time training and scaling (Section 4.3). Finally, our approach is **orthogonal** to advanced outcome-based RL methods [1]; since these methods are agnostic to our framework, they can be seamlessly integrated, potentially yielding further gains.
> > >
> > > We will include these training time comparisons and discussion points in the revised paper.
> > >
> > > [1] DAPO: An Open-Source LLM Reinforcement Learning System at Scale
> > >
> > > We hope these additional experiments and discussions fully address your remaining concerns. If our response has satisfactorily resolved your questions, we kindly ask that you consider raising your score. Thank you again for your time.

---

### Official Review · Reviewer_Tf33 · 2026-03-12

**Soundness:** 3
**Presentation:** 3
**Significance:** 3
**Originality:** 3
**Overall Recommendation:** 4
**Confidence:** 3

**Summary:**

This paper proposes rePIRL, an inverse RL-inspired framework that learns effective PRMs with minimal assumptions about expert policies. rePIRL involves a dual learning process that alternates between policy and PRM updates, along with techniques tailored to scaling inverse RL for LLMs. Theoretically, the proposed rePIRL framework can unify online and offline PRM learning under additional assumptions. Extensive experiments on math and coding reasoning benchmarks demonstrate the effectiveness of the proposed method and the utility of the trained PRM.

**Compliance With Llm Reviewing Policy:**

Affirmed.

**Final Justification:**

My concerns have been addressed, and I tend to maintain my current positive scores.

**Key Questions For Authors:**

1.	In figure 1(b), how many responses are generated for majority voting? What is the generation temperature?

**Limitations:**

See weakness

**Strengths And Weaknesses:**

Strength

1.	rePIRL can unify online and offline PRM learning under additional assumptions.

2.	Experiments demonstrate the clear advantage of rePIRL over baseline methods and the utility of the trained PRM in various scenarios.


Weakness

1.	This paper adopts greedy decoding for evaluation. As discussed in [1,2], greedy decoding for reasoning models results in higher repetition rates and significant variability. For reasoning models, it’s more common to adopt pass@1 metric averaged over multiple responses in the literature[3,4,5] , with non-greedy decoding.

2. Compared with outcome-based RL methods, the proposed method needs to train an additional reward model of a similar size to the policy model, which leads to higher computation. Besides, how will the proposed method perform, compared with more advanced outcome-based RL methods for LRMs such as [3,4] ?

[1] https://arxiv.org/abs/2501.12948
[2] https://huggingface.co/Qwen/Qwen3-4B
[3] https://arxiv.org/abs/2505.12366
[4] https://arxiv.org/abs/2509.22611
[5] https://arxiv.org/abs/2506.01347

---

> ### Author Rebuttal · Authors · 2026-03-30
>
> > This paper only adopts the greedy decoding for evaluation.
>
> **Response:**
> Thanks for pointing this out and for providing the relevant references [1-5]. We agree that while greedy decoding provides a deterministic, low-compute baseline, pass@1 with non-greedy decoding offers a more robust assessment of a model's true reasoning capabilities. To address this, we conduct additional evaluations comparing rePIRL against two strong baselines (RLOO and PRIME) using non-greedy decoding (temperature = 0.6, k=16 samples per prompt), and report the expected pass@1 on our math benchmarks. The results are summarized below:
>
> | Qwen2.5-3B-Instruct | MATH500 | AIME 2024 | MinervaMath | AMC | OlympiadBench | Average |
> |---|---|---|---|---|---|---|
>  | Base | 0.417 | 0.047 | 0.207 | 0.275 | 0.266 | 0.242 |
>  | RLOO | 0.595 | 0.060 | 0.247 | 0.320 | 0.286 | 0.301 |
>  | PRIME | 0.541 | 0.079 | 0.239 | 0.319 | 0.278 | 0.291 |
> | rePIRL | **0.589** | **0.075** | **0.266** | **0.332** | **0.282** | **0.308** |
>
> | Qwen3-4B-Base | MATH500 | AIME 2024 | MinervaMath | AMC | OlympiadBench | Average |
> |---|---|---|---|---|---|---|
> | Base | 0.495 | 0.058 | 0.156 | 0.271 | 0.268 | 0.249 |
> | RLOO | 0.686 | 0.102 | 0.270 | 0.427 | 0.384 | 0.373 |
> | PRIME | 0.667 | 0.108 | 0.320 | 0.414 | 0.369 | 0.375 |
> | rePIRL | **0.697** | **0.121** | **0.290** | **0.432** | **0.390** | **0.385** |
>
>
> As shown in the table, while the absolute performance numbers shift as expected under non-greedy decoding, the relative performance gains of our proposed method remain **significant and consistent** with our original findings. In particular, rePIRL continues to outperform the strongest baselines within each model family, achieving average improvements of 0.7% over RLOO on Qwen2.5-3B-Instruct and 1.0% over PRIME on Qwen3-4B-Base. We will include these non-greedy evaluations and add the given citations.
>
> > The proposed method has higher computational overhead due to the additional reward model.
>
> **Response:**
> We appreciate the reviewer’s observation. Although rePIRL introduces an additional reward model, it does not incur substantial computational overhead compared to many state-of-the-art methods. In particular, its model and computational complexity are comparable to approaches that rely on PRMs, such as PRIME and Math-Shepherd (Section 5). Our reward model does not need to be the same size as the policy and can be a small model.
>
> When compared directly to outcome-based RL baselines like RLOO, rePIRL introduces only a ~1.4x training time overhead (e.g., 13 hours vs. 9 hours to fine-tune Qwen2.5-3B-Instruct on math tasks). Importantly, this limited overhead is offset by several key advantages. First, rePIRL provides a unified framework (Section 3.3), which empirically leads to consistent performance gains over all baselines across diverse tasks and model architectures. Second, the learned PRM serves as a reusable and standalone module, benefiting a number of downstream applications such as test-time training and scaling (Section 4.3).
> > How does the method compare with advanced outcome-based RL methods like [3,4]?
>
> **Response:**
> We thank the reviewer for highlighting these advanced outcome-based RL baselines. To address this, we have conducted an additional comparison against QAE [4] on our math benchmarks using the Qwen2.5-3B-Instruct model. Specifically, we follow the settings detailed in [4] (e.g., DAPO with quantile advantage estimation) and report the expected pass@1 accuracy to evaluate the models' reasoning capabilities. The results are shown below:
> | Qwen2.5-3B-Instruct  | MATH500 | AIME 2024 | MinervaMath | AMC | OlympiadBench | Average |
> |---|---|---|---|---|---|---|
>  | QAE | 0.492 | 0.06 | 0.224 | 0.29 | 0.271 | 0.267 |
> | rePIRL | **0.589** | **0.075** | **0.266** | **0.332** | **0.282** | **0.308** |
>
> The result shows that our method consistently outperforms QAE, achieving an absolute average improvement of 4.1% across the math benchmarks. This further validates the effectiveness of our PRM-guided approach compared to advanced outcome-based RL techniques.
>
> Furthermore, our approach is orthogonal to these advanced outcome-based RL methods. Since they are agnostic to our framework, they can be readily integrated, potentially leading to further performance gains.
> >  In figure 1(b), how many responses are generated for majority voting? What is the generation temperature?
>
> **Response:**
> We thank the reviewer for pointing this out. To clarify, the x-axis in Figure 1(b) denotes the number of responses generated for test time scaling; specifically, we evaluated settings using 4, 32, and 64 responses. The generation temperature was set to 0.8, with additional inference details provided in Appendix B.

---

> > ### Author Rebuttal · Reviewer_Tf33 · 2026-04-02
> >
> > Thanks for the authors' efforts. My concerns have been addressed, and I tend to maintain my current scores.

---

### Decision · Program_Chairs · 2026-04-30

**Decision:**

Accept (regular)

**Comment:**

This paper proposes rePIRL, an inverse RL-inspired framework for learning process reward models from expert trajectories without requiring token-level annotations or access to the expert's reward function. The method alternates between policy and PRM updates, and the authors show it theoretically unifies online and offline PRM learning. Experiments on math and coding benchmarks demonstrate consistent improvements over baselines including RLOO, PRIME, and MCTS.

The theoretical contribution of unifying existing PRM learning methods under a common framework is valuable, and the empirical results are solid across multiple model families and benchmarks. The learned PRM demonstrates standalone utility in test-time training, test-time scaling, and hard problem bootstrapping. During rebuttal, the authors provided more experiment results which resolved 3 reviewers' concerns. However, reviewer 6fKw's concern about computational overhead versus outcome-based methods is a genuine practical consideration.

Recommendation: Weak Accept